# PromptFix: You Prompt and We Fix the Photo

**Yongsheng Yu**[1][*]   **Ziyun Zeng**[1][*]   **Hang Hua**[1]   **Jianlong Fu**[2]   **Jiebo Luo**[1]

[1]University of Rochester, [2]Microsoft Research

{yyu90,zzeng24}@ur.rochester.edu, {hhua2,jluo}@cs.rochester.edu, jianf@microsoft.com

## Abstract

Diffusion models equipped with language models demonstrate excellent controllability in image generation tasks, allowing image processing to adhere to human instructions. However, the lack of diverse instruction-following data hampers the development of models that effectively recognize and execute user-customized instructions, particularly in low-level tasks. Moreover, the stochastic nature of the diffusion process leads to deficiencies in image generation or editing tasks that require the detailed preservation of the generated images. To address these limitations, we propose PromptFix, a comprehensive framework that enables diffusion models to follow human instructions to perform a wide variety of image-processing tasks. First, we construct a large-scale instruction-following dataset that covers comprehensive image-processing tasks, including low-level tasks, image editing, and object creation. Next, we propose a high-frequency guidance sampling method to explicitly control the denoising process and preserve high-frequency details in unprocessed areas. Finally, we design an auxiliary prompting adapter, utilizing Vision-Language Models (VLMs) to enhance text prompts and improve the model's task generalization. Experimental results show that PromptFix outperforms previous methods in various image-processing tasks. Our proposed model also achieves comparable inference efficiency with these baseline models and exhibits superior zero-shot capabilities in blind restoration and combination tasks. The dataset and code are available at `https://www.yongshengyu.com/PromptFix-Page`.

## 1 Introduction

In recent years, diffusion models [19, 57, 66] have achieved remarkable advancements in text-to-image generation. Benefiting from large-scale training on image-text pairs [59], these models can generate highly realistic and diverse images that align with text prompts. They have been successfully applied to various real-world applications, including visual design, photography, digital art, and the film industry. In addition, models trained with instruction-following data [7] have shown promising results in understanding human instruction and performing the corresponding image-processing tasks. Previous studies [21–23, 74, 73] have illustrated that with instruction-following data, we can simply fine-tune a text-to-image generation model to perform various vision tasks such as image editing [7, 21, 74], object detection [22], segmentation [23], inpainting [23, 73], and depth estimation [22, 9]. To follow the success of these methods, we train our model utilizing input-goal-instruction triplet data for low-level image-processing tasks.

We first overcome the challenge of lacking the instruction-following data for low-level tasks. Specifically, we collect image pairs by generating degraded images from the source images and adopting data from existing datasets. Then, we employ GPT4 [51] to generate the diverse text instructions for each task. We obtain $\sim 1.01$ million input-goal-instruction triplets in the collected dataset. This dataset covers various low-level tasks including image inpainting, object creation, image dehazing, colorization, super-resolution, low light enhancement [8], snow removal, and watermark removal. We enrich the dataset through back-translation augmentation by swapping the inpainted and origi-

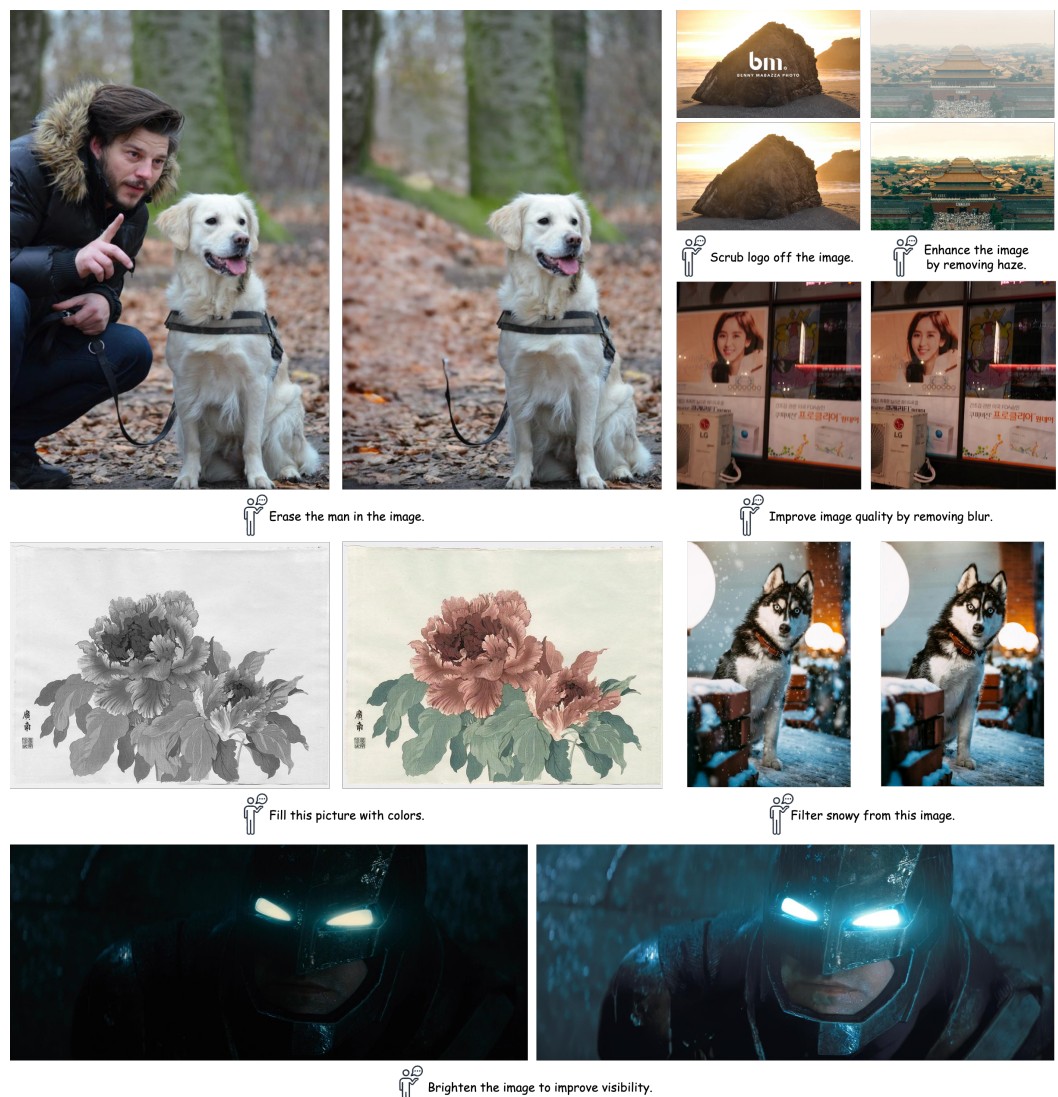

Figure 1: We propose PromptFix, a unified diffusion model capable of performing multiple image-processing tasks. It can understand user-customized editing instructions and perform the corresponding tasks with high quality. One of the key advantages of PromptFix is high-frequency information preservation, ensuring that image details are maintained throughout VAE decoding. PromptFix can handle various images with different aspect ratios.

nal images within the triplet and inverting the semantic orientation of the prompt. This technique effortlessly converts datasets from object removal to object creation. We also provide comprehensive details in Section 3.

With the dataset, we design a new diffusion-based model named PromptFix that can understand user-customized instructions and perform the corresponding low-level image-processing tasks. In PromptFix, we address several challenges that compromise the performance of the model. First, the use of stable-diffusion architecture [57] as a generative prior often faces the issue of spatial information loss, which is caused by VAE compression [20]. Unlike unconditional or text-to-image generation, maintaining spatial detail consistency in image processing poses significant challenges, particularly with high-frequency components like text, as shown in Figure 5. To tackle this problem, we introduce High-frequency Guidance Sampling, in which we use a low-pass filter operator [50, 53] to calculate the *fidelity constraint* and integrate VAE skip-connect features during inference with a lightweight LoRA [27] fusion. Second, since the generative prior is not trained on low-level images, so relying solely on instructions may not always yield the desired outcome, especially when the image degradation is severe. To tackle this degradation adaptation problem, we introduce an *auxiliary prompt*

*module* to provide models with more descriptive text prompts to enhance controllability for image generation. The auxiliary text prompt can be obtained by VLMs [43]. This approach introduces a semantic caption for a degraded image and the description of its defects, such as blurriness or insufficient lighting. The *auxiliary prompt module* is implemented by an additional attention layer in diffusion U-Net that adapts both instruction and *auxiliary prompts* as conditions and intermittently omits instructional prompts during training. We identify three key advantages of this approach: 1) enabling the model to process images with severe degradation, such as extremely low-resolution images, 2) adapting the model for blind restoration for different types of image degradation, and 3) providing additional pathways for a more precise semantic representation of the target image.

Experimental results demonstrate that our model achieves superior performance in the instruction-based paradigm across three image editing tasks (colorization, watermark removal, object removal) and four image restoration tasks (dehazing, desnowing, super-resolution, and low-light enhancement) in terms of perceptual pixel similarity [75] and no-reference image quality [72]. In summary, our contributions are three-fold:

- We propose a comprehensive dataset tailored for seven image processing tasks. The dataset contains $\sim 1.01$ million diverse paired input-output images along with corresponding image editing instructions.

- We propose a new all-in-one instruction-guided diffusion model – PromptFix for low-level image-processing tasks. Extensive experimental results show that PromptFix outperforms previous methods in a wide variety of image-processing tasks and exhibits superior zero-shot capabilities in blind restoration and combination tasks.

- We introduce two approaches – high-frequency guidance sampling and auxiliary prompt module to diffusion models to effectively address the issues of high-frequency information loss and the failure in processing the severe image degradation for instruction-based diffusion models in low-level tasks.

## 2 Related Work

### 2.1 Instruction-guided Image Editing

Instruction-guided image editing significantly improves the ease and precision of visual manipulations by adhering to human directions. In traditional image editing, models primarily focused on singular tasks such as style transfer or domain adaptation [24, 55], leveraging various techniques to encode images into a manipulatable latent space, such as those used by StyleGAN [34]. Concurrently, the advent of text-to-image diffusion models [26, 57, 63, 65] has broadened the scope of image editing [7, 25]. Kim et al. [35] showed how to perform global changes, whereas Avrahami et al. [4] successfully performed local manipulations using user-provided masks for guidance. While most works that require only text (i.e., no masks) are limited to global editing [17, 36]. Bar-Tal et al. [6] proposed a text-based localized editing technique without using any mask, showing impressive results. For local image editing, precise manipulations are possible by inpainting designated areas using either user-provided or algorithmically predicted masks [16], all while preserving the visual integrity of the adjacent areas. In contrast, instruction-based image editing operates through direct commands like "add fireworks to the sky," avoiding the need for detailed descriptions or regional masks. Recent approaches utilize synthetic input-goal-instruction triples [7] and incorporate human feedback [76] to execute editing instructions effectively. Despite the advances in using diffusion models for various instruction-guided image editing tasks, there is still a notable gap in research specifically addressing instruction-guided image restoration with these models. Our study aims to bridge this gap by collecting a comprehensive dataset of paired low-level instruction-driven image editing examples and proposing an all-in-one model for low-level tasks and editing.

### 2.2 Large Language Models for Vision

Recent advancements in the development of Large Language Models (LLMs) have led to the emergence of powerful models with extensive capabilities [15, 31, 39, 43, 77]. These LLMs, pre-trained on large-scale internet-based datasets, are equipped with broad knowledge bases that enhance their zero-shot and in-context learning abilities [5, 29, 51]. Furthermore, there is a growing focus on using LLMs for multimodal tasks [3, 30, 42, 45], incorporating methods like vision-language

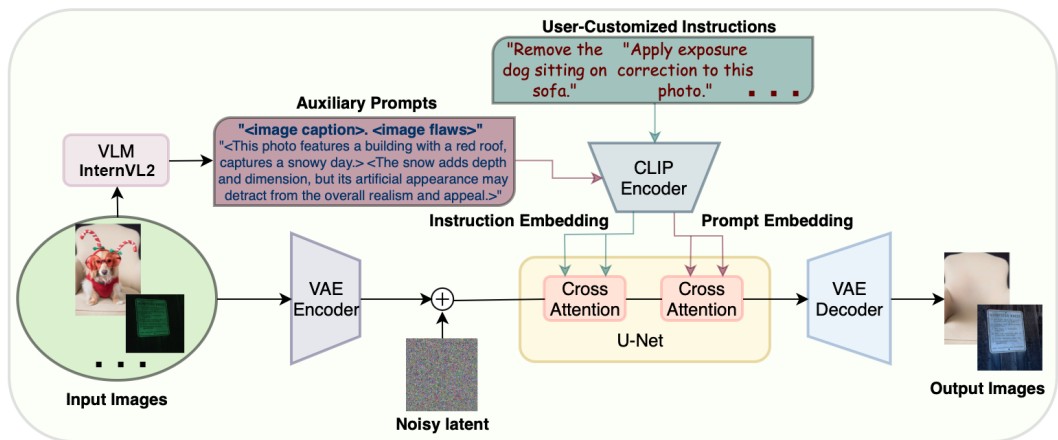

Figure 2: The architecture of our proposed PromptFix.

alignment and adapter fine-tuning. These techniques ensure that the visual data processed by visual encoders is semantically aligned with the textual input of LLMs [43]. This approach has spurred significant advancements in Text-to-Image generation, leading to the development of various LLM-based diffusion models for these tasks [28, 40, 41]. Despite these successes, there remains a relative scarcity of research focusing on using large Vision-Language Models (VLMs) for instructed image editing, particularly in detailed, low-level editing tasks.

## 3 Data Curation

Current off-the-shelf image datasets [7, 74, 76] with instructional annotations primarily facilitate image editing research, encompassing tasks such as color transfer, object replacement, object removal, background alteration, and style transfer. Nevertheless, their overlap with low-level applications is limited. Moreover, we find it challenging to achieve satisfactory results for existing models in image restoration. Our goal is to construct a comprehensive visual instruction-following dataset specifically for low-level tasks. We obtain $\sim 1.01$ million training triplet instances.

**Paired Image Collection.** We initially gather source images from various existing datasets. Subsequently, we produce degraded and inpainted images to create an extensive set of paired image data. We compile approximately two million raw data points across eight tasks: image inpainting, object creation, image dehazing, image colorization, super-resolution, low-light enhancement, snow removal, and watermark removal. For the test set, we randomly select 300 image pairs for each task. More details about the dataset composition are provided in the Appendix A.1.

**Instruction Prompts Generation.** For each low-level task, we utilized GPT-4 to generate diverse training instruction prompts $\mathcal{P}_{\text{instruction}}$. These prompts include task-specific and general instructions. The task-specific prompts, exceeding 250 entries, clearly define the task objectives. For example, "Improve the visibility of the image by reducing haze" for dehazing. The general instructions include five ambiguous commands that we retained as "negative" prompts to promote adaptive tasks. The specific instruction prompts used for training are detailed in the appendix. For watermark removal, super-resolution, dehazing, snow removal, low-light enhancement, and colorization tasks, we also generate "auxiliary prompts" for each instance. These auxiliary prompts describe the quality issues for the input image and provide semantic captions. More details are discussed in Section 4.2.

## 4 Methodology

Let $I \in \mathbb{R}^{H \times W \times 3}$ denote the degraded input image. Our PromptFix model aims to enhance $I$ using the prompt $\mathcal{P}$ and the diffusion model $\mathcal{H}$.

---

**Algorithm 1** High-frequency Guidance Sampling.

---

    **Input:** $\mathcal{H}, D_\theta, I, \mathcal{P}$

    **Hyper-parameter:** $\lambda, \{\sigma_t\}_{t=1}^T, \{\alpha_t\}_{t=1}^T, S_{\text{churn}}, S_{\text{noise}}, S_{\text{tmin}}, S_{\text{tmax}}$

1: **sample** $\mathbf{z}_T \sim \mathcal{N}(\mathbf{0}, \sigma_T^2 \mathbf{I})$

2: **for** $t \in \{T, \dots, 1\}$ **do**                $\triangleright \gamma_t = \begin{cases} \min\left(\frac{S_{\text{churn}}}{N}, \sqrt{2}-1\right) & \text{if } \sigma_t \in [S_{\text{tmin}}, S_{\text{tmax}}] \\ 0 & \text{otherwise} \end{cases}$

3:     **sample** $\boldsymbol{\epsilon}_t \sim \mathcal{N}\left(\mathbf{0}, S_{\text{noise}}^2 \mathbf{I}\right)$

4:     $\hat{\mathbf{z}}_t \leftarrow \mathbf{z}_t + \sqrt{\hat{\sigma}_t^2 - \sigma_t^2}\boldsymbol{\epsilon}_t, \hat{\sigma}_t \leftarrow \sigma_t + \gamma_t \sigma_t$   $\triangleright$ Increase noise temporarily and inject new noise for state transition.

5:     $\hat{\boldsymbol{\epsilon}}(\mathbf{z}_t, t), \hat{\mathbf{z}}_{t-1} \leftarrow \mathcal{H}(\hat{\mathbf{z}}_t, I, \mathcal{P})$       $\triangleright$ Return predicted noise from neural network and denoised latent.

6:     $\mathbf{z}_{t-1} \leftarrow \hat{\mathbf{z}}_t + (\sigma_{t-1} - \hat{\sigma}_t)(\hat{\mathbf{z}}_t - \hat{\mathbf{z}}_{t-1})/\hat{\sigma}_t$      $\triangleright$ Execute Euler step moving forward from $\hat{\sigma}_t$ to $\sigma_{t-1}$.

7:     $\mathbf{z}_{t \to 0} \leftarrow (\mathbf{z}_t - \sigma_t \hat{\boldsymbol{\epsilon}}(\mathbf{z}_t, t))/\alpha_t$                    $\triangleright$ Equation (3)

8:     $\theta \leftarrow \theta - e^{-\lambda t}\nabla_\theta \mathcal{L}(I, D_\theta(\mathbf{z}_{t \to 0}))$

9: **end for**

---

## 4.1 Diffusion Model

Diffusion models transform data into noise through gradual Gaussian perturbations during a forward process and subsequently reconstruct samples from this noise in a backward process. In the forward phase, an original data point, denoted as $\mathbf{z}_0$, is incrementally altered towards a Gaussian noise distribution $\boldsymbol{\epsilon} \sim \mathcal{N}(0, \mathbf{I})$, according to the equation:

$$\mathbf{z}_t = q(\mathbf{z}_0, \boldsymbol{\epsilon}, t) = \alpha_t \mathbf{z}_0 + \sigma_t \boldsymbol{\epsilon}, \quad \forall t \in [0, T], \tag{1}$$

where $\alpha_t$ and $\sigma_t$ are coefficients that manage the signal-to-noise ratio at each interpolation point $\mathbf{z}_t$. This process aims to maintain variance, adopting coefficient strategies as detailed in sources such as [33]. Modeled as a stochastic differential equation (SDE) in continuous time, the forward process can be expressed as $d\mathbf{z} = \mathbf{f}(\mathbf{z}, t)dt + g(t)d\mathbf{w_t}$, where $\mathbf{f}(\mathbf{z}, t)$ is a vector-valued drift coefficient, $g(t)$ is the diffusion coefficient, and $\mathbf{w_t}$ represents Brownian motion at time $t$.

The backward diffusion process, made possible by notable characteristics of the SDE, is rearticulated via Fokker-Planck dynamics [66] to yield deterministic transitions with consistent probability densities, creating the *probability flow ODE*:

$$d\mathbf{z} = \left[\mathbf{f}(\mathbf{z}, t) - \frac{1}{2}g(t)^2 \nabla_\mathbf{z} \log p_t(\mathbf{z})\right] dt. \tag{2}$$

This equation outlines a transport mechanism that is learnable through maximum likelihood techniques, applying the perturbation kernel of diffused data samples $\nabla_\mathbf{z} \log p_t(\mathbf{z}|\mathbf{z}_0)$, as demonstrated in [32, 66]. Next, we sample $\mathbf{z}_t \sim \mathcal{N}(0, I)$ to initialize the probability flow ODE, and the estimates for the score function via $\hat{\boldsymbol{\epsilon}}(\mathbf{z}_t, t)/\sigma_t$. We employ the Euler method [64, 66] among numerical ODE solvers to obtain the solution trajectory: $\mathbf{z}_0 \approx \mathcal{H}_N \circ \mathcal{H}_{N-1} \circ \cdots \circ \mathcal{H}_1(\mathbf{z}_T)$, where $\mathcal{H}$ denotes the diffusion model and $N$ represents the neural function evaluations (NFEs) for sampling.

During the training phase, a simple diffusion loss [26] is utilized, whereby the neural network still employs forward inference to predict noise. The sample data estimate $\mathbf{z}_0$ can be obtained at any step $t$ by using the current noisy data and the predicted noise and is derived as:

$$\mathbf{z}_{t \to 0} = \frac{\mathbf{z}_t - \sigma_t \hat{\boldsymbol{\epsilon}}(\mathbf{z}_t, t)}{\alpha_t}. \tag{3}$$

To reduce computational costs, the aforementioned diffusion process initiates from isotropic Gaussian noise samples in the latent space [57], rather than the pixel space. This space transformation is facilitated through VAE compression [20]. The VAE autoencoder comprises an encoder $E(\cdot)$ and a left inverse decoder $D(\cdot)$. For instance, an image $x$ can be encoded into a latent code $E(x)$, which can then be approximately reconstructed back into the pixel space as $x \approx D \circ E(x)$.

## 4.2 VLM-based Auxiliary Prompt Module

Given that low-level image processing focuses on handling degraded images rather than real-world images, we adopt the integration of a VLM to estimate an *auxiliary prompt* for the low-level image $I$. This auxiliary prompt encompasses both semantic captions and defect descriptions to enhance the semantic clarity of the target image, thereby addressing the instructional gaps inherent in low-level image processing tasks.

Based on text dialogues parameterized by $\omega$ within a VLM $\mathcal{V}(\cdot; \omega)$, we employ a frozen VLM, specifically the InternVL2 [15] model, which integrates visual and linguistic modalities as inputs. We facilitate this model to receive paired degraded image $I$ and a textual query $\mathcal{Q}$. To handle the visual input, the InternVL2 first employs a pre-trained encoding model to map each modality into a shared representation space. The visual encoding model $\phi$ embeds $I$ into the textual space, resulting in $\phi(I)$, which is then combined with the tokenized language embedding $\tau(\mathcal{Q})$. These combined embeddings are fed into the large language model, producing the textual response $\mathcal{R}$.

$$\mathcal{R}(I, \mathcal{Q}) = \mathcal{V}(\phi(I), \tau(\mathcal{Q}); \omega) \tag{4}$$

The visual encoding model of InternVL2 has not undergone extensive fine-tuning in the degradation domain. To acquire an explicit understanding from both semantic and low-level defect perspectives, we meticulously curate the queries $\mathcal{Q}_{\text{semantic}}$ and $\mathcal{Q}_{\text{defect}}$ to guide InternVL2, respectively. Specific query instances are provided in the Appendix. As illustrated in Figure 4 and described by Equation 5, we concatenate the responses related to semantics and degradation textually, forming $\mathcal{P}_{\text{auxiliary}}$, which serves as the *auxiliary prompt*. This acts as a supplement to the instruction prompt $\mathcal{P}_{\text{instruction}}$.

$$\mathcal{P}_{\text{auxiliary}} = [\mathcal{R}(I, \mathcal{Q}_{\text{semantic}}), \mathcal{R}(I, \mathcal{Q}_{\text{defect}})] \tag{5}$$

The conditioned text prompts guide the diffusion model by injecting the embedding into the cross-attention layer [57]. After obtaining the auxiliary prompt, a straightforward approach involves concatenating it with the user-input instruction prompt before feeding the text embedding into the diffusion model. However, this concatenation can render the entire prompt excessively long, leading to forced truncation during tokenization. Therefore, after utilizing the pre-trained CLIP visual encoder ViT-L/14 [54] to extract linguistic features, we process the text embeddings of $\mathcal{P}_{\text{instruction}}$ and $\mathcal{P}_{\text{auxiliary}}$ separately. We introduce additional cross-attention layers identical to the original ones, as depicted in Figure 2. The embeddings of $\mathcal{P}_{\text{instruction}}$ and $\mathcal{P}_{\text{auxiliary}}$ are fed into the Key and Value heads of consecutive attention networks, respectively, thereby achieving an augmented cross-attend adaptation.

### 4.3 High-frequency Guidance Sampling

There is a fundamental requirement in image restoration and generation tasks: the processed image must maintain high accuracy in semantics. We observe that vanilla VAE reconstructions tend to lose image details such as textual rendering, which contains high-frequency information, as shown in Figure 5. Therefore, we propose high-frequency guidance sampling to balance the quality and fidelity of generation.

The denoising sampling is based on the EDM formulation [33]. To maintain spatial information, we utilize a modified VAE Decoder $D_\theta$ to map from the latent space to the pixel space. We modify the VAE decoder by passing skip-connect features from the VAE encoder through additional LoRA convolutions [27] to merge the feature map. The LoRA networks are initialized randomly, with their trainable parameters denoted as $\theta$. Since the parameters of the LoRA convolution are lightweight, merely multi-step backpropagation can maintain high-frequency consistency without requiring extensive fine-tuning.

We propose a *fidelity constraint* to model the spatial discrepancy between the image and the ground truth. We implement two types of high-pass operators to extract high-frequency signals from the degraded image. For the Fourier filtering operator $\mathcal{F}(\cdot)$, we convert the generated image from the spatial to the frequency domain using the Discrete Fourier Transform [50]. High-frequency components are then isolated via high-pass filtering and reconstituted into an image through the Inverse Fourier Transform [50]. Meanwhile, we apply the Sobel [53] edge detection operator $\mathcal{S}(\cdot)$ as a complement. The fidelity constraint evaluates the divergence between the high-frequency components of the ground truth and the processed image, ensuring the preservation of spatial information throughout the sampling process. Additionally, to obtain the image at time step $t$, we utilize predicted noise $\epsilon$ from diffusion model $\mathcal{H}$ to compute the $\mathbf{z}_{t \to 0}$ estimation at any time step. The fidelity constraint is calculated as follows:

$$\mathcal{L}(I, D_\theta(\mathbf{z}_{t \to 0})) = \|\mathcal{F}(I) - \mathcal{F}(D_\theta(\mathbf{z}_{t \to 0}))\|_2^2 + \|\mathcal{S}(I) - \mathcal{S}(D_\theta(\mathbf{z}_{t \to 0}))\|_2^2 \tag{6}$$

Given that $\mathbf{z}_{t > 0}$ represents a noisy latent variable, assigning equal weight to each timestep's latent is impractical. To mitigate the cumulative error induced by this practice, we introduce a time-scale weight $e^{-\lambda t}$. The overall sampling algorithm is detailed in Algorithm 1.

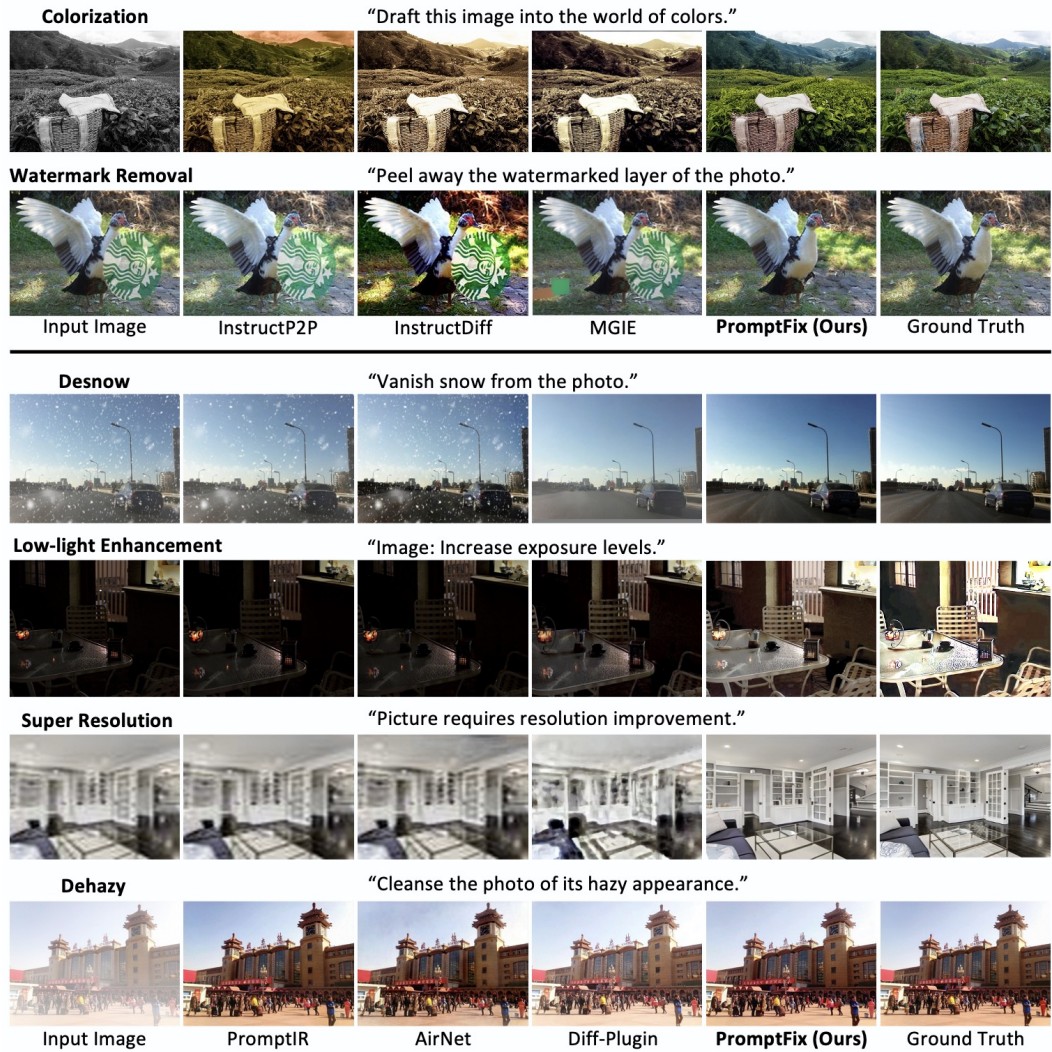

Figure 3: Qualitative comparison between PromptFix and other instruct-driven diffusion methods (InstructP2P [7], InstructDiff [23], and MGIE [21]) for image processing, as well as low-level generalist techniques (PromptIR [52], AirNet [38], and Diff-Plugin [46]) for image restoration.

## 5 Experiments

### 5.1 Experimental Setup

**Implementation details**. We train PromptFix for 46 epochs on 32 NVIDIA V100 GPUs, employing a learning rate of $1 \times 10^{-4}$ with the Adam optimizer. The training input resolution is set to $512 \times 512$, matching the capabilities of our backbone models, InternVL2 [15] and Stable Diffusion 1.5 [57]. To facilitate classifier-free guidance [7], we randomly drop the input image latent, instruction, and auxiliary prompt with a probability of 0.075 during training. The hyperparameter $\lambda$ for the time-scale weight in Algorithm 1 is empirically set to 0.001. For more implementation details, please refer to the appendix.

**Baselines and metrics**. We adopt instruction-based generalist models, such as InstructP2P [7], MGIE [21], and InstructDiffusion [23], as our primary comparison. MGIE employs VLM-guided techniques for image editing, while InstructDiffusion addresses overlapping tasks with our training objectives, including watermark removal and inpainting. Additionally, we evaluate all-in-one image restoration methods like AirNet [38] and PromptIR [52] (which do not support instruction input), as well as image restoration expert models, fine-tuned for specific sub-tasks [46, 73]. We assess the

Table 1: Quantitative comparison is conducted across seven low-level datasets with a $512 \times 512$ input resolution. Expert models refer to approaches, such as Diff-plugin [46], which use non-generalizable training pipelines and maintain separate pre-trained weights for each of the four restoration tasks. Image Restoration Generalist Methods denote models that integrate multiple low-level tasks into a single framework. Instruct-driven Diffusion Methods represent diffusion generative baselines that follow human language instructions. ↑ indicates higher is better and ↓ indicates lower is better. The **best** and second best results are in bold and underlined, respectively.

| Method | LPIPS↓ / ManIQA↑ | | | LPIPS↓ / ManIQA↑ | | | |
| --- | --- | --- | --- | --- | --- | --- | --- |
| | Colorization | Object Removal | Watermark Removal | Low-light Enhance. | Desnow | Dehazy | Super Res. |
| Expert Models | | | | | | | |
| Diff-Plugin [46] | - | - | - | 0.227/0.453 | 0.133/0.508 | 0.033/0.758 | 0.097/0.555 |
| Inst-Inpaint [73] | - | 0.227/0.593 | - | - | - | - | - |
| Image Restoration Generalist Methods | | | | | | | |
| PromptIR [52] | - | - | - | 0.330/0.539 | 0.235/0.553 | 0.037/0.764 | 0.105/0.442 |
| AirNet [38] | - | - | - | 0.332/0.541 | 0.245/0.589 | 0.039/0.780 | 0.107/0.450 |
| Instruction-driven Diffusion Methods | | | | | | | |
| InstructP2P [7] | 0.394/0.424 | 0.177/0.791 | 0.341/0.378 | 0.581/**0.460** | 0.365/**0.560** | 0.216/0.625 | 0.234/0.528 |
| InstructDiff [23] | 0.433/0.256 | 0.071/**0.811** | 0.247/0.675 | 0.368/0.309 | 0.255/0.530 | 0.124/0.711 | 0.233/0.623 |
| MGIE [21] | 0.425/0.393 | - | 0.463/0.506 | 0.491/0.356 | 0.483/0.417 | 0.249/0.704 | 0.397/0.385 |
| **PromptFix (ours)** | **0.233/0.489** | **0.054**/0.810 | **0.127/0.750** | **0.135**/0.423 | **0.103**/0.535 | **0.088/0.752** | **0.143/0.642** |

Figure 4: Qualitative analysis of VLM-guided blind restoration for desnowing, dehazing, and low-light enhancement. The results are obtained from PromptFix without explicit task instructions, relying solely on the input image. The auxiliary prompt, automatically generated by a VLM during inference, includes semantic captions and defect descriptions, indicated by <blue> and <yellow> tags, respectively.

similarity of the generated images to the ground truth using metrics such as PSNR, SSIM [69], and LPIPS [75]. For no-reference image quality evaluation, we utilize the ManIQA [72] metric.

## 5.2 Quantitative and Qualitative Results

Table 1 illustrates the comparative analysis of image restoration and editing techniques, evaluated via LPIPS and ManIQA metrics. The expert model – Diff-Plugin shows limited but notable performance in low-light enhancement (LPIPS/ManIQA: 0.227/0.453) and desnowing (0.133/0.508). Among generalist methods, AirNet demonstrates balanced capabilities in tasks like desnowing and dehazing, achieving LPIPS/ManIQA scores of 0.245/0.589 and 0.039/0.780, respectively. However, the instruction-driven diffusion methods reveal a more nuanced picture, with PromptFix emerging as particularly promising. It excels in colorization (LPIPS/ManIQA: 0.233/0.489), object removal (0.054/0.810), and watermark removal (0.071/0.811), consistently outperforming others. InstructP2P and InstructDiff also perform well in specific tasks, such as low-light enhancement and dehazing, but do not match the overall versatility of PromptFix. MGIE, though effective in certain domains, lacks the consistency seen in "PromptFix (Ours)." This highlights the robustness and superior performance

Table 2: Quantitative comparison of multi-task processing on 200 sampled test images, each paired with a degraded version exhibiting three defects (e.g., desnowing, dehazing and super-resolution) and the corresponding ground truth.

| Method | PSNR↑ | SSIM↑ | LPIPS↓ | ManIQA↑ |
|---|---|---|---|---|
| PromptIR [52] | 19.30 | 0.7385 | 0.2282 | 0.4310 |
| AirNet [38] | 19.15 | 0.7197 | 0.2359 | 0.4576 |
| InstructP2P [7] | 14.19 | 0.5845 | 0.3046 | 0.3790 |
| InstructDiff [23] | 17.12 | 0.6387 | 0.2865 | 0.4365 |
| **PromptFix** | **22.05** | **0.7654** | **0.1519** | **0.4889** |

Table 3: Quantitative comparison on three low-level tasks. PromptFix † denotes blind restoration without input instruction prompts. The compared baselines specify task objectives explicitly.

| Method | LPIPS↓ / ManIQA↑ | | |
|---|---|---|---|
| | Low-light Enhancement | Desnow | Dehazy |
| PromptIR [52] | 0.331 / 0.539 | 0.236 / 0.533 | 0.038 / 0.764 |
| AirNet [38] | 0.332 / 0.541 | 0.245 / 0.589 | 0.039 / 0.781 |
| InstructP2P [7] | 0.581 / 0.461 | 0.365 / 0.560 | 0.216 / 0.626 |
| InstructDiff [23] | 0.369 / 0.309 | 0.256 / 0.531 | 0.124 / 0.712 |
| PromptFix† | 0.161 / 0.413 | 0.115 / 0.531 | 0.148 / 0.755 |

of PromptFix across diverse image processing tasks and indicates the potential of PromptFix to set new benchmarks in the field, driven by advanced instruction-driven diffusion methodologies.

Figure 3 illustrates the visual comparison across all selected baseline models. In colorization, our PromptFix produces the most visually accurate and vibrant results, closely resembling the ground truth. For the watermark removal task, it effectively restores the original image without introducing artifacts, outperforming MGIE [21] and other methods. In desnowing and low-light enhancement, PromptFix achieves clearer and more natural outputs, significantly reducing noise and enhancing visibility. Additionally, in super-resolution, PromptFix demonstrates remarkable clarity and accuracy, preserving fine details and surpassing all comparative methods. In dehazing, although PromptFix's performance is visually comparable to image restoration experts PromptIR [52] and AirNet [38], PromptFix outperforms the recent stable-diffusion-based method Diff-plugin [46], achieving a clean, sharp appearance, and closely matching the ground truth.

### 5.3 Ablation Study

**Effectiveness of High-frequency Guidance Sampling**. The High-frequency Guidance Sampling (HGS) method is introduced to balance *fidelity* and *quality*. To validate the effectiveness of HGS, we conduct qualitative experiments and quantitative experiments. As depicted in Figure 5, in a low-light scenario, the model aims to enhance the visibility (*quality*) of the input image while preserving its original textual details (*fidelity*). For baselines leveraging stable-diffusion as a generative prior, the strong compression capability of VAE also brings the issue of spatial information loss, as demonstrated by InstructDiff [23], MGIE [21], and Diff-Plugin [46] in Figure 5. This issue is independent of the model's ability to effectively follow instructions. As shown by the variant "Ours w/o HGS," our method significantly enhances low-light images compared to the three baselines, yet still fails to retain small-scale textual structures. By incorporating HGS, as seen in "Ours," the proposed framework delivers a high-fidelity solution that also meets the low-light enhancement instruction. The usage of $\mathcal{F}(\cdot)$ and $\mathcal{S}(\cdot)$ improves the quality of the image generated, demonstrated by the quantitative results shown in Table 4.

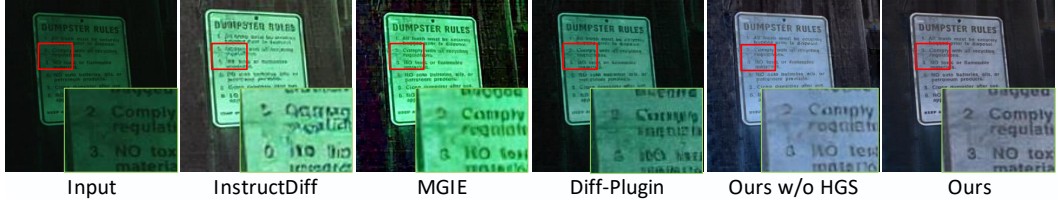

| Input | InstructDiff | MGIE | Diff-Plugin | Ours w/o HGS | Ours |

Figure 5: Preservation of low-level image details using the proposed High-frequency Guidance Sampling (HGS) method, compared to previous VAE-based baselines [21, 23, 46] utilizing stable-diffusion architecture.

**VLM-guided blind restoration**. We utilize InternVL2 [15] to generate auxiliary prompts and leave the instruction prompt empty. This approach enables users to input an image without the need to provide instructions for its restoration. We evaluate the model's performance on such blind restoration tasks, including low-light enhancement, desnowing, and dehazing. As shown in Table 3, our model achieves performance comparable to four baselines, showing minimal perceptual differences from the ground truth and superior zero-shot capabilities.

Table 4: Quantitative study on HGS and Auxiliary Prompting. $\mathcal{F}(\cdot)$ and $\mathcal{S}(\cdot)$ represent the type of high-frequency operators used in HGS to guide sampling.

| HGS ($\mathcal{F}(\cdot)$) | HGS ($\mathcal{S}(\cdot)$) | Auxiliary Prompting | LPIPS↓ | ManIQA↑ |
|---|---|---|---|---|
| - | - | ✓ | 0.2068 | 0.6487 |
| - | ✓ | ✓ | 0.1707 | 0.6300 |
| ✓ | - | ✓ | 0.1795 | 0.6195 |
| ✓ | ✓ | - | 0.1990 | 0.5856 |
| ✓ | ✓ | ✓ | 0.1600 | 0.6274 |

Table 5: Ablation Study on Different Types of Instruction Prompt. $\mathbb{A}$: instructions used during training; $\mathbb{B}$: out-of-training human instructions with fewer than 20 words; $\mathbb{C}$: out-of-training human instructions with 40-70 words.

| Method | Instruction Prompt Type | LPIPS↓ | ManIQA↑ |
|---|---|---|---|
| PromptFix | $\mathbb{A}$ | 0.1600 | 0.6274 |
| | $\mathbb{B}$ | 0.1639 | 0.6258 |
| | $\mathbb{C}$ | 0.1823 | 0.5958 |

**Multi-task processing**. Although PromptFix is not explicitly trained to handle multiple low-level tasks simultaneously within the same image, it demonstrates the capability for multi-task processing. We construct the validation dataset with 200 images, and each image contains 3 restoration tasks such as colorization, watermark removal, low-light enhancement, desnowing, dehazing, and super-resolution. We benchmarked PromptFix against AirNet and PromptIR, both generalist image restoration methods, as well as InstructP2P and InstructDiff, which are instruction-driven diffusion methods. As shown in Table 2, PromptFix outperforms these baselines, achieving superior image quality, structural similarity, and minimal perceptual differences from the ground truth, as evidenced by competitive PSNR, SSIM, and LPIPS scores, along with a higher ManIQA score indicating visually pleasing and high-quality results. Conversely, while methods like InstructP2P and InstructDiff perform well in specific metrics, they do not match the overall balanced performance of PromptFix. These results indicate the robustness and versatility of PromptFix.

**Different types of instruction prompts**. We verify PromptFix's generalization to various human instructions in Table 5, by conducting ablation comparisons with three types of prompts: instructions used during training and out-of-training human instructions with fewer than 20 words and with 40-70 words. PromptFix's performance slightly declines with out-of-training instructions, but the change is negligible. It indicates that PromptFix is robust for instructions under 20 words, which is generally sufficient for low-level processing tasks. We observe a performance drop with longer instructions, possibly due to the long-tail effect of instruction length in the training data. Although low-level processing tasks usually don't require long instructions, addressing this issue by augmenting the dataset with longer instructions could be a direction for future work.

# 6    Conclusion

We present PromptFix, a novel diffusion-based model along with a large-scale visual-instruction training dataset, designed to benefit instruction-guided low-level image processing. PromptFix effectively addresses challenges related to spatial information loss and degradation adaptation by high-frequency guidance sampling and a VLM-based auxiliary prompt module. These mechanisms improve the model's performance in the instruction-based image-processing paradigm. Extensive experiment results demonstrate PromptFix's advanced capabilities of generating accurate and high quality images. In addition to the improvement in terms of conventional metrics, we observe that PromptFix is also effective at processing multi-task processing and achieving blind restoration in low-light enhancement, desnowing, and dehazing.

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

# A  Appendix

## A.1  PromptFix Dataset

The PromptFix dataset contains 1,013,320 triplet instances among 7 different tasks containing object removal, image dehazing, colorization, debluring, low light enhancement, snow removal and watermark removal. Each instance contains an input image, a processed image, an instruction, and an auxiliary prompt generated by InternVL2 [15] which is only available for tasks except object removal. The dataset is available at `https://huggingface.co/datasets/yeates/PromptfixData`.

**Object removal**: Mulan [67] provides multi-layer image decomposition annotations, inpainting occluded areas and isolating instances in a scene on separate RGBA layers. We combine the background layer and $n$ foreground object layers to obtain the ground truth image and the background layer and $n + 1$ foreground object layers to obtain the input image, where $n \in \mathbb{N}$. Different object layers will be placed onto the background image based on their foreground-background relationships.

**Image dehazing**: We use a combination of synthetic datasets (RESIDE [37], SRRS [13]) and real-world datasets (Dense-Haze [1], O-Haze [2]) for training, comprising 100 pairs of real-world images and 102,230 pairs of indoor and outdoor synthetic images from ITS [37], OTS [37], SOTS outdoor [37], and nyuhaze500 [37].

**Colorization**: We utilize a subset of Laion-5b [60] and Flickr comprising 317,225 images and generate grayscale images.

**Deblurring**: We use a combination of GoPro [49], HIDE [61], LFDOF [58], RealBlur [56], TextOCR [62], Wider-Face [71] and a subset of CelebA [48], which collectively contain 70,600 pairs of images.

**Low light enhancement**: We use a combination of LOL [70], SID [11], SMID [10], SDSD [68], and construct a set of low light synthetic data based on Pexels and Flickr by reducing brightness and adding noise, which collectively contains 212,618 pairs of images.

**Snow removal**: We use a combination of SRRS [13], CSD [14], RVSD [12], and Snow100K [47], which collectively contains 99,177 pairs of images.

**Watermark removal**: We use a combination of CLWD [44] and LOGO30K [18], which collectively contain 124,805 pairs of images.

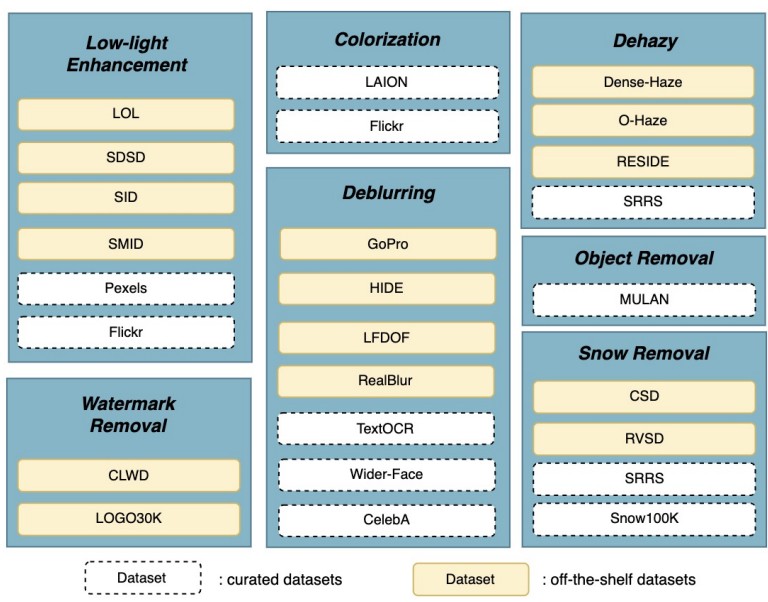

Figure 6: Data composition.

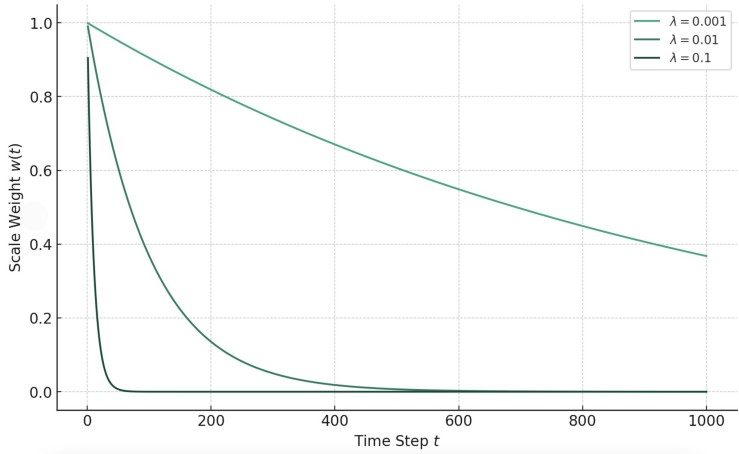

Figure 7: Exponential Decay Weight Functions.

## A.2 Limitations

Without relying on user input instructions, our PromptFix achieves blind restoration for low-level enhancement, dehazing, and desnowing. However, we observe that this approach occasionally results in out-of-conditioned image control, where the model performs text-to-image generation based on the auxiliary prompt rather than image processing. Although blind restoration is a feature of our VLM-based Auxiliary Prompt Module, for known degradations, we recommend providing user-custom instructions to specify the restoration.

High-frequency Guidance Sampling significantly helps preserve the original image details, counteracting the spatial information loss caused by VAE compression. Nevertheless, we find that adopting HGS makes the restored image slightly resemble the degraded image. While PromptFix remains promising relative to baselines, as shown in Figure 5, results without HGS appear brighter than those with HGS. As we have consistently claimed, employing HGS involves a trade-off between fidelity and quality. In scenarios where VAE reconstruction is sufficiently faithful and user needs prioritize quality, HGS may be omitted.

## A.3 More Implementation Details

Below are the more detailed implementation specifics:

- $\mathcal{Q}_{\text{semantic}}$: *"Describe this image and its style in a very detailed manner"*
- $\mathcal{Q}_{\text{defect}}$: *"Introduce the drawback of the image"*

**Parameters of the Sobel Operator**: This involves applying two $3 \times 3$ kernels:

$$G_x = \begin{bmatrix} -1 & 0 & 1 \\ -2 & 0 & 2 \\ -1 & 0 & 1 \end{bmatrix} \quad \text{and} \quad G_y = \begin{bmatrix} -1 & -2 & -1 \\ 0 & 0 & 0 \\ 1 & 2 & 1 \end{bmatrix}$$

These kernels calculate gradients in the horizontal and vertical directions, respectively. By convolving these kernels with the image, we obtain the gradient magnitudes at each pixel. The default parameters include a mid-range intensity threshold to discern significant edges.

**Time-scale Weight** $\lambda$: This weight is determined based on the time step of the diffusion model. When the time step is larger, there is more noise, and the fidelity constraint should be decayed. As depicted in Figure 7, the impact of the high-frequency loss exponentially decays with increasing timestep. Empirically, we have set $\lambda$ to 0.001.

**Weight of Fidelity Constraint**: The time-scale weight, which is determined by the time step of the diffusion model, decreases as the time step increases, due to the increase in noise. Consequently, the fidelity constraint should be decayed accordingly.

**HGS Algorithm Notion Clarification**: Using $T$ to represent the neural function evaluations (NFEs) for sampling, and $S$ to denote a series of hyperparameters:

- $S_{\text{churn}}$ controls the level of stochasticity.
- $S_{\text{noise}}$ adjusts the noise standard deviation.
- $S_{\text{tmin}}$ and $S_{\text{tmax}}$ define the range of noise levels for stochasticity.

## A.4 Efficiency analysis

We conduct an efficiency analysis for PromptFix and other recent baseline models, including Diff-plugin [46] and InstructDiff [23], by profiling the floating-point operations (FLOPs) of their respective diffusion models. As demonstrated in the table below, our method exhibits comparable efficiency to these advanced models. This analysis reveals the capability of PromptFix to maintain high computational efficiency while delivering superior performance in various image-processing tasks.

|                    | TFLOPs |
|--------------------|--------|
| Diff-Plugin [46]   | 1.1    |
| InstructDiff [23]  | 0.8    |
| PromptFix (Ours)   | 0.8    |

## A.5 More Results

**General PromptFix Results**. We present additional image processing results in Figures 9-15, encompassing tasks including watermark removal, colorization, low-light enhancement, super-resolution, dehazing, desnowing, inpainting, and object removal via bounding boxes. These results are generated by our PromptFix model, utilizing an all-in-one pre-trained weight and guided by user-customized instructions.

**Mutli-task processing and blind restoration**. In Figure 8, we present the qualitative results of multi-task processing as a complement to the quantitative assessments detailed in Table 2. Additionally, we provide further visualizations of VLM-guided blind restoration outcomes as in Figure 16.

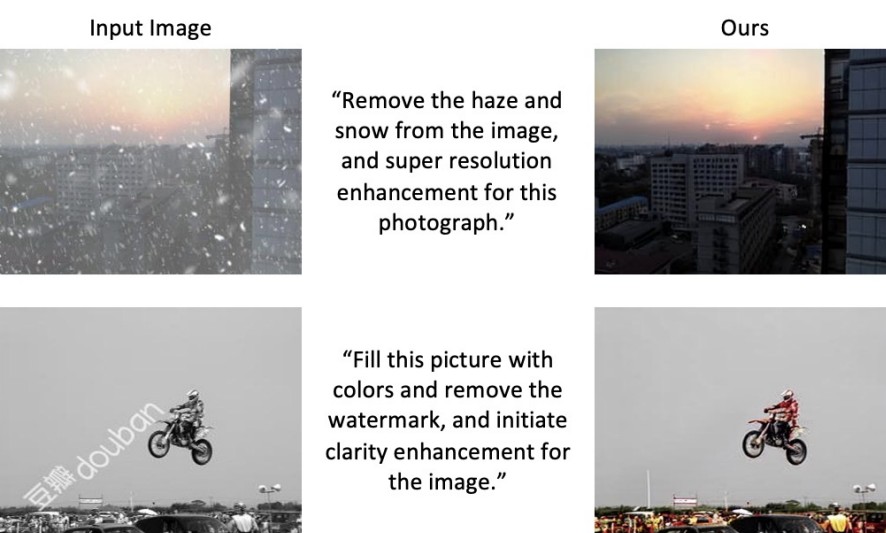

Figure 8: Visual results for Mutli-task processing.

"Eliminate watermark on the image."

Input Image

Ours

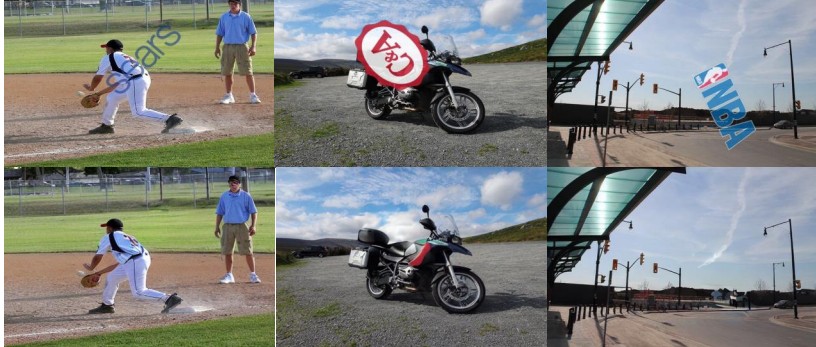

"Image: Wipe out watermark."

Input Image

Ours

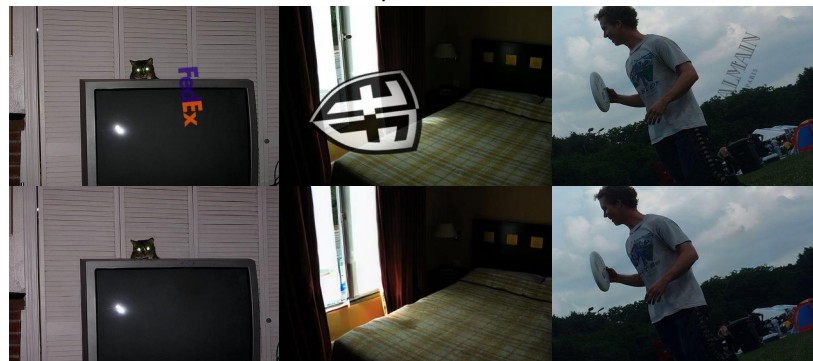

"Cleanse this photo of watermark."

Input Image

Ours

"Scrub logo off the image."

Input Image

Ours

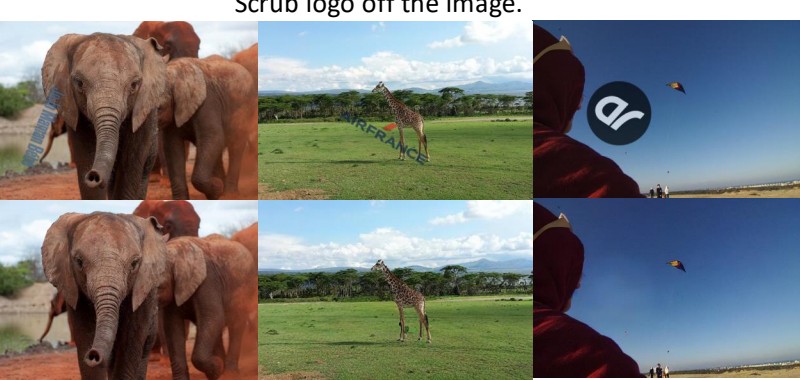

Figure 9: More results for watermark removal.

"Apply a chromatic touch to this photograph."

Input Image

Ours

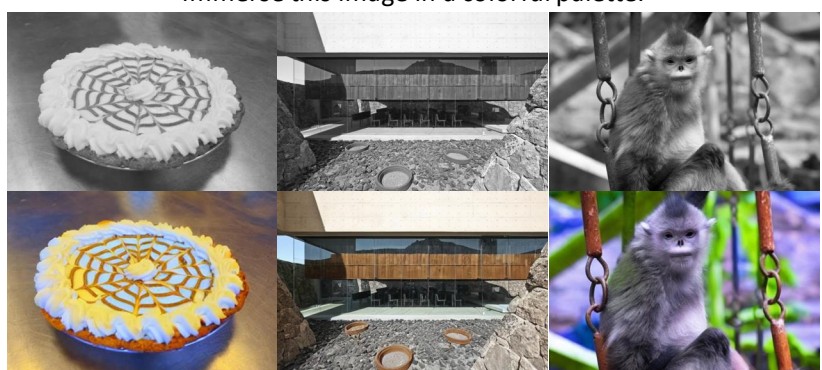

"Immerse this image in a colorful palette."

Input Image

Ours

"Colorize the image to enhance its visual appeal."

Input Image

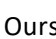

Ours

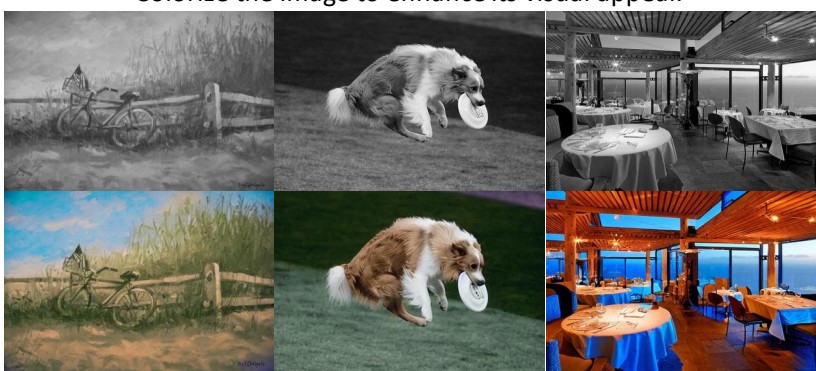

"Paint the image with color tones."

Input Image

Ours

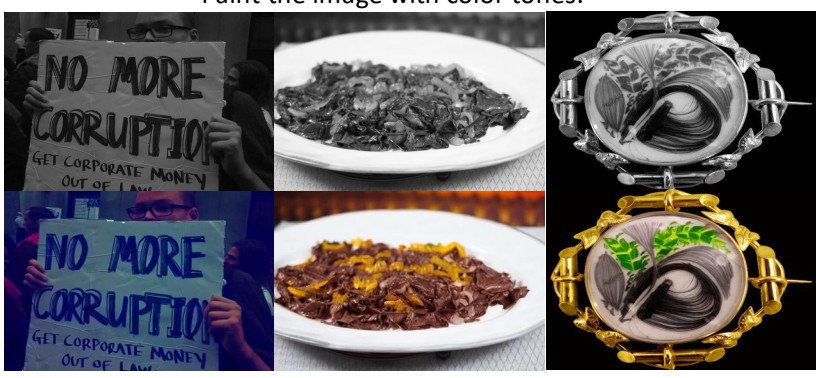

Figure 10: More results for image colorization.

"Execute illumination boost on the picture."

Input Image

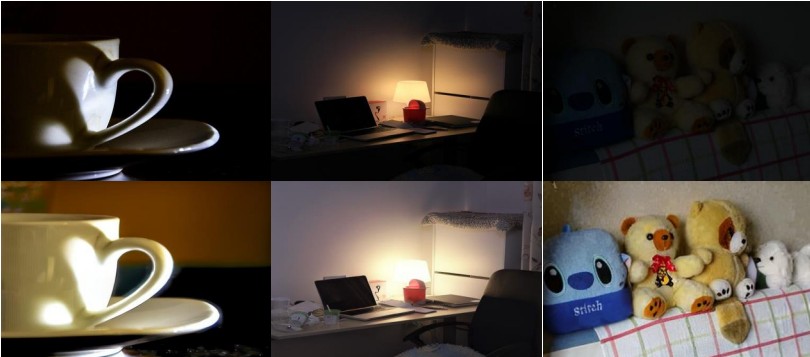

Ours

"Activate detail enhancement in dark areas for this photograph."

Input Image

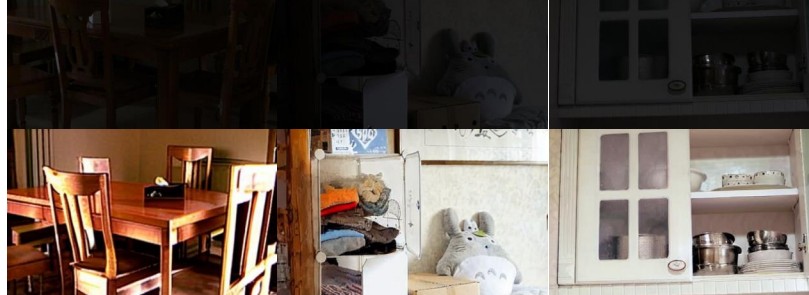

Ours

"Picture: Enhance for better night visibility."

Input Image

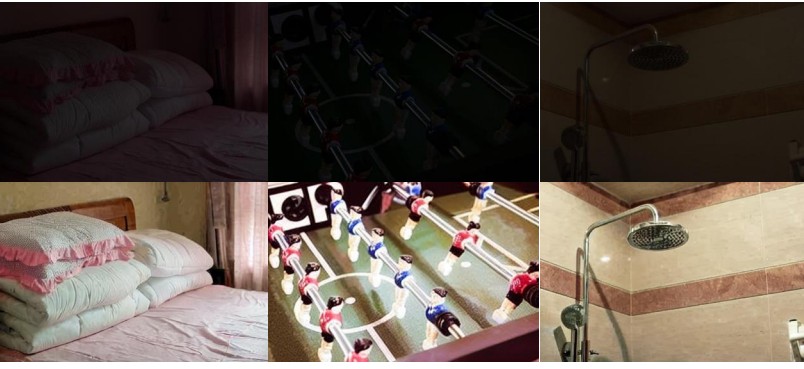

Ours

"Enhance overall lighting of the image."

Input Image

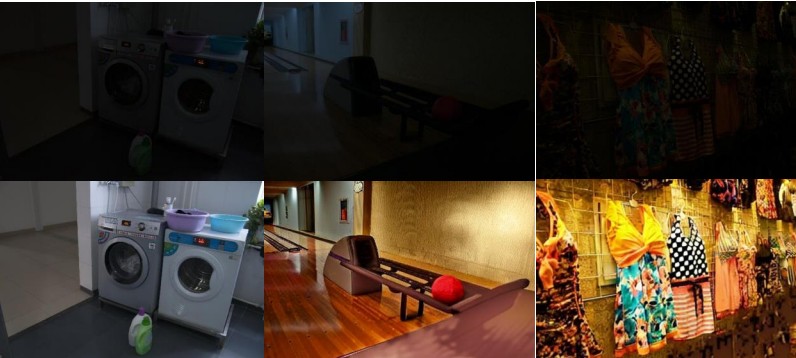

Ours

Figure 11: More results for low-light enhancement.

"Remove the haze from this image."

Input Image

Ours

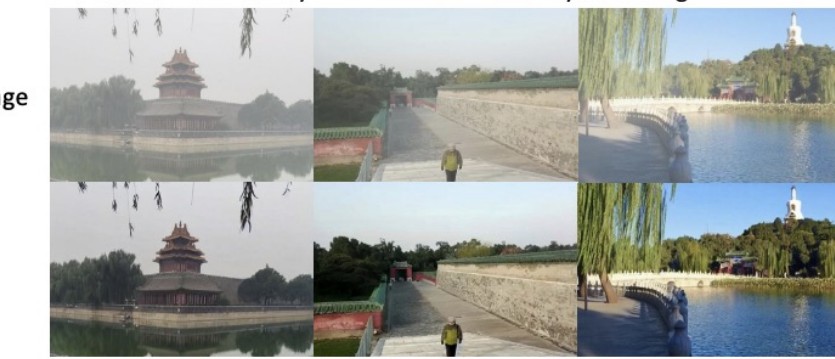

"Picture: Clarify the obscured details by removing haze."

Input Image

Ours

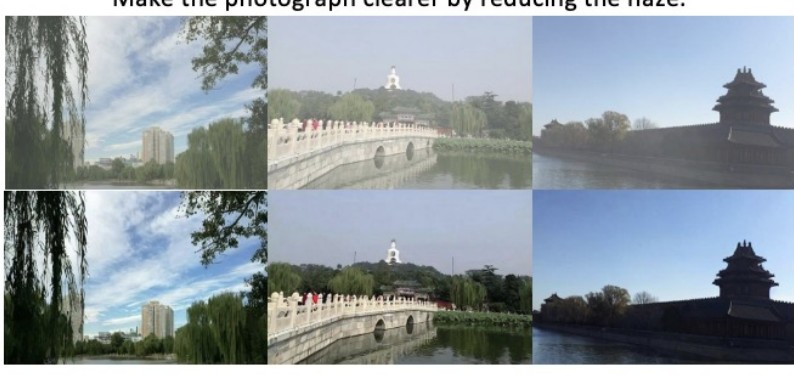

"Make the photograph clearer by reducing the haze."

Input Image

Ours

"Reduce the haziness to sharpen picture."

Input Image

Ours

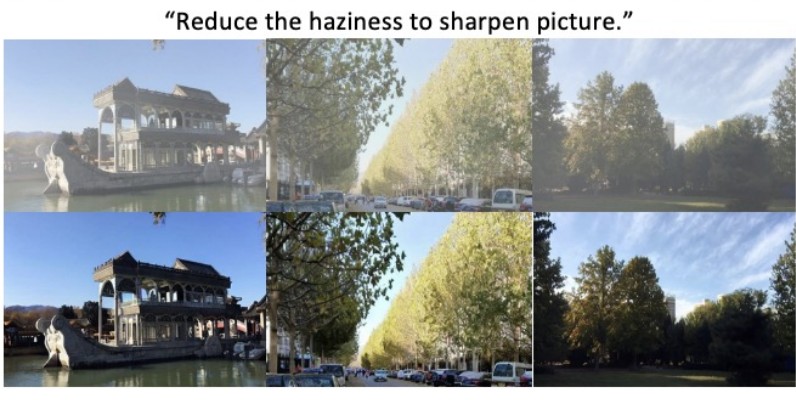

Figure 12: More results for image dehazing.

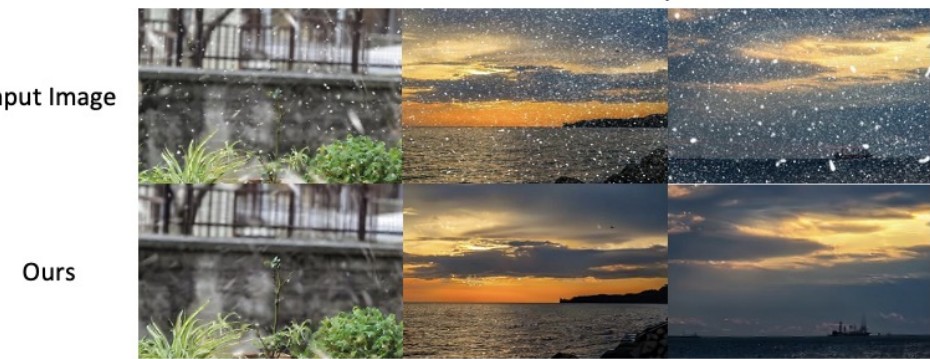

Figure 13: More results for desnowing.

Input Image                                                                 Ours

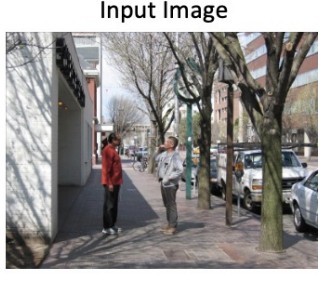    "Remove the man    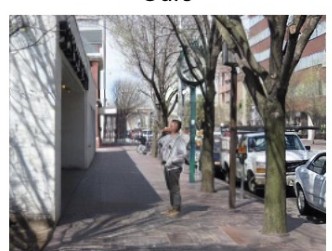
                        which is to the right of
                        the tall brick building."

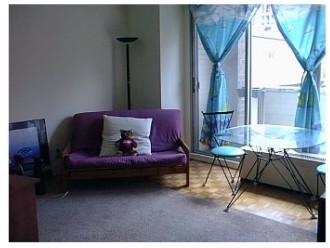    "Remove the stuffed    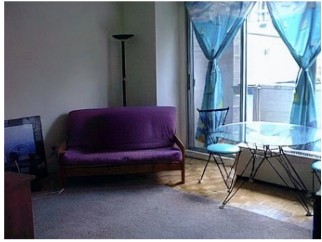
                        teddy bear which is on
                        the purple cloth wood
                        couch."

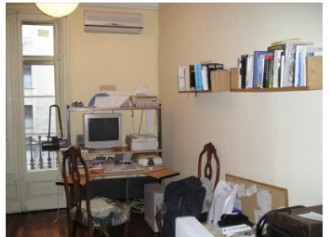    "Remove the gray off    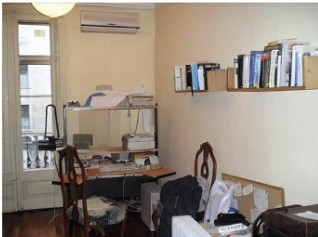
                        computer monitor which
                        is on the brown desk."

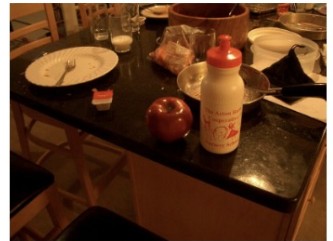    "Remove the metal empty    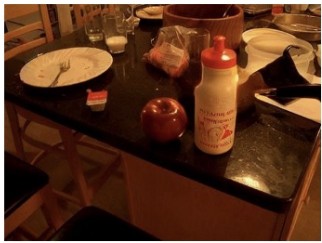
                        pan which is to the right
                        of the brown large wood
                        round empty bowl."

Figure 14: More results for image inpainting.

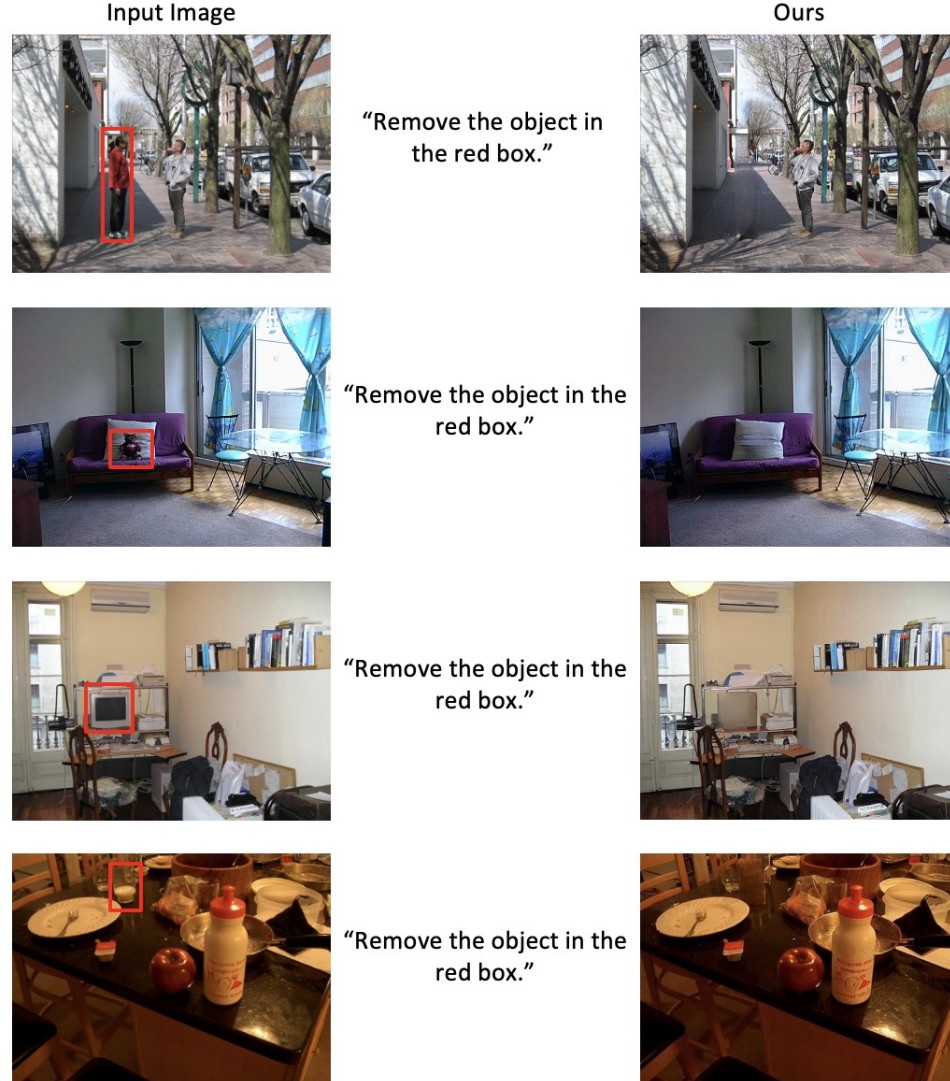

Figure 15: More results for object removal by a bounding box.

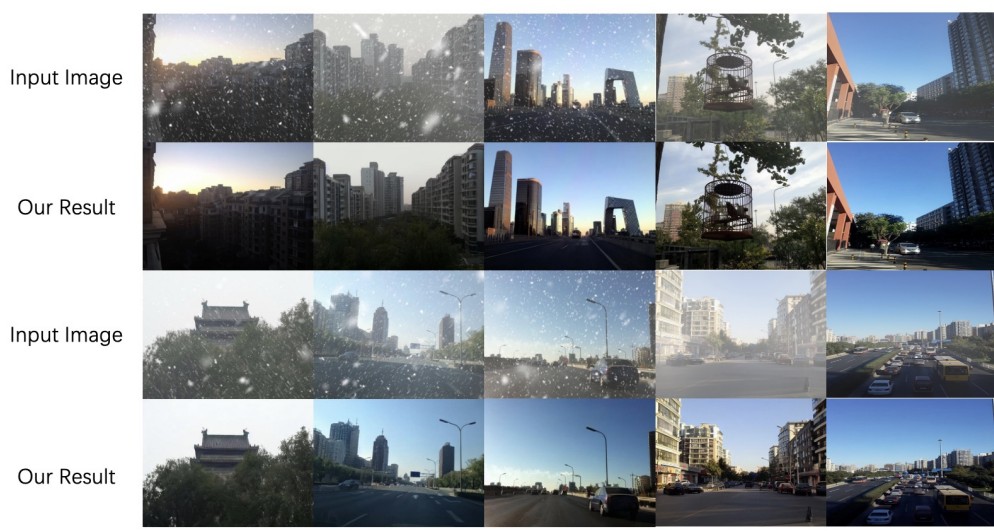

Figure 16: Visual results for VLM-guided blind restoration.

