# OpenReview forum: "PromptFix: You Prompt and We Fix the Photo"
_NeurIPS.cc/2024/Conference — NeurIPS 2024 poster_

### Official Review · Reviewer_6nKQ · 2024-07-01

**Soundness:** 3
**Presentation:** 3
**Contribution:** 2
**Rating:** 6
**Confidence:** 4

**Summary:**

The paper introduces PromptFix, a novel framework that significantly enhances the capability of diffusion models in following human instructions for a wide range of image-processing tasks. The authors propose a comprehensive multi-modal dataset and further design a frequency-based diffusion model trained on this dataset. Experiments show competitive performance on various restoration tasks.

**Strengths:**

1. The large-scale, instruction-following dataset that covers comprehensive image-processing tasks could be helpful in the field of low-level image processing.
2. Integrating a high-frequency guidance sampling method and an auxiliary prompting adapter shows reasonable problem-solving capability.

**Weaknesses:**

1. While instructions are necessary for users, the types of degradation tasks (such as snow removal and low-light enhancement) are clearly defined. In other words, for images with the same type of degradation (such as foggy images), how to choose different instructions to achieve the best results remains to be clarified. Additionally, it should be considered whether users can use instructions other than the task-specific prompts provided by the authors.
2. The modules used are relatively common. The AuxiliaryPrompt, serving as an information cue derived from the image itself, has been utilized in the literature as referenced in [1,2]. And High-frequency Guidance is also a commonly used method [3,4], albeit in this paper, it is constrained in the form of a loss.

Ref:

[1] Lin J, Zhang Z, Wei Y, et al. Improving image restoration through removing degradations in textual representations. CVPR 2024.

[2] Chiu M T, Zhou Y, Zhang L, et al. Brush2Prompt: Contextual Prompt Generator for Object Inpainting. CVPR 2024.

[3] Miao Y, Deng J, Han J. WaveFace: Authentic Face Restoration with Efficient Frequency Recovery.  CVPR 2024.

[4] Zhao C, Cai W, Dong C, et al. Wavelet-based Fourier information interaction with frequency diffusion adjustment for underwater image restoration. CVPR 2024.

**Questions:**

1. It is important to consider whether the comparative methods have been retrained or fine-tuned on the same dataset. If not, the fairness of the comparison is questionable.
2. In line 205 of the paper, "low-pass operators" might be incorrect; it should probably be "high-pass operators".
3. The test set was randomly selected, and its domain is similar to that of the training set. It is necessary to supplement the comparison with untrained standard datasets.
4. The paper lacks ablation experiments for auxiliary prompts. Additionally, the High-frequency Guidance method lacks numerical metrics and only presents visual results.
5. In Table 1, for the unified restoration task, both PromptIR and AirNet, despite not incorporating instructions, demonstrate good performance, with half of the metrics surpassing those of PromptFix. This seemingly limits the perceived superiority of the proposed method's performance.
6. In line 228, The MGIE[19] was published in ICLR 2024. It has been incorrectly marked in the references.

**Limitations:**

As shown in the Weaknesses and Questions.

---

> ### Author Rebuttal · Authors · 2024-08-07
>
> We sincerely appreciate your invaluable feedback and the opportunity to address your queries regarding our approach.
>
> > **Q1**: While instructions are necessary for users, the types of degradation tasks (such as snow removal and low-light enhancement) are clearly defined. In other words, for images with the same type of degradation (such as foggy images), how to choose different instructions to achieve the best results remains to be clarified. Additionally, it should be considered whether users can use instructions other than the task-specific prompts provided by the authors.
>
> PromptFix is designed to understand and follow user-customized instructions for low-level image processing tasks. We discuss this in detail in **General Response III**. As illustrated in the table in General Response III, the impact of different instructions on numerical performance is minimal when the prompt length is less than 20, demonstrating that PromptFix has low sensitivity to varying instructions.
>
>
> > **Q2**: The modules used are relatively common. The AuxiliaryPrompt, serving as an information cue derived from the image itself, has been utilized in the literature as referenced in [1,2]. High-frequency Guidance is also a commonly used method [3,4], albeit in this paper, it is constrained in the form of a loss.
>
> - Integrating VLM to improve restoration results is a straightforward method. However, our paper goes beyond [1,2] by discovering and analyzing that the VLM-based auxiliary prompt module helps the diffusion model handle multi-degradation processing and blind restoration.
> - Unlike existing high-frequency modules [3,4], our HGS is training-free and adaptable to multiple low-level tasks, not limited to a specific restoration domain.
>
>
> > **Q3**: It is important to consider whether the comparative methods have been retrained or fine-tuned on the same dataset. If not, the fairness of the comparison is questionable.
>
> Thank you for your advice. We discuss this in the **General Response - IV**, please refer to it.
>
>
> > **Q4**: In line 205 of the paper, "low-pass operators" might be incorrect; it should probably be "high-pass operators".
>
> Thank you for pointing out. We will revise it.
>
> > **Q5**: The test set was randomly selected, and its domain is similar to that of the training set. It is necessary to supplement the comparison with untrained standard datasets.
>
> Good suggestion. We provide more real-world low-level processing results in the updated PDF. Please refer to it.
>
> > **Q6**: The paper lacks ablation experiments for auxiliary prompts. Additionally, the High-frequency Guidance method lacks numerical metrics and only presents visual results.
>
>
> We provide the quantitative ablation experiments in **General Response - II**. Please refer to it.
>
> > **Q7**: In Table 1, for the unified restoration task, both PromptIR and AirNet, despite not incorporating instructions, demonstrate good performance, with half of the metrics surpassing those of PromptFix. This seemingly limits the perceived superiority of the proposed method's performance.
>
> It is challenging for an all-in-one model to outperform all the task-expert models in all metrics. However, our model offers several advantages:
>
> 1. **Natural Language Guidance:** PromptFix is guided by user-customized instructions, rather than instruction tags, enabling it to perform tasks like object removal, which PromptIR and AirNet cannot achieve.
> 2. **Task Unification:** Our single model unifies various image processing tasks, such as colorization, object removal, and watermark removal.
> 3. **Comprehensive Restoration:** PromptFix can perform blind restoration and handle multi-degradation processing.
>
>
> > **Q8**: In line 228, The MGIE[19] was published in ICLR 2024. It has been incorrectly marked in the references.
>
> Thanks for pointing out. We will revise it.

---

> > ### Comment · Reviewer_6nKQ · 2024-08-11
> >
> > Thanks for the response. The explanation is generally reasonable.
> >
> > For General Response-II, the results show that HGS and Auxiliary Prompting are effective.
> >
> > For General Response-IV, the fine-tuning of InstructDiff demonstrates the effectiveness of the proposed method.
> >
> > But I still have some concerns and will keep my score.
> >
> > For General Response-I, please confirm that test results on untrained real-world data have been provided. This will demonstrate the method's generalization performance.
> >
> > For General Response-III, it is recommended to provide examples of the instructions designed in B & C settings, and explain how the GPT-4 generated instructions are guided by the prompts.
> >
> > For Q2, HGS is a loss that does not require training, and each input image needs to fine-tune the decoder during the sampling process.
> > Is this design reasonable? Please explain why the decoder is updated instead of updating xt or the U-Net.
> >
> > For Q7, AuxiliaryPrompt provides supplementary information about the image, which is also input through the image encoder, although it is not displayed in text form.
> > As for High-frequency Guidance, some high-frequency information in real-world degraded images contains noise, making this guidance somewhat rough.
> > Additionally, in the final steps of the diffusion model, it could have a negative impact.
> > Therefore, these two contributions are somewhat common.

---

> > > ### Author Response · Authors · 2024-08-11
> > >
> > > We sincerely appreciate your feedback and acknowledgment of the validity of our experiments in General Responses II and IV, which demonstrate the effectiveness of our proposed method. We would also like to address the concerns raised in your comments:
> > >
> > > 1. **Regarding the Concern on General Response - I Data**
> > >
> > >    We confirm that none of the data used originates from the training set. The data sources include online platforms, such as watermarked images from Adobe Stock and Shutterstock, black-and-white photography from Pexels, dark scenes from films, and personal mobile photography. This approach aims to validate the generalization capability of our method using real-world data, rather than relying on simulated data created by degrading specific images.
> > >
> > > 2. **On Providing Examples for General Response - III's Instructions**
> > >
> > >    We show some examples of instructions from General Response III. Due to space constraints, we provide one example for each task:
> > >
> > >    |  | $\mathbb{B}$ | $\mathbb{C}$ |
> > >    |:---:| :---| :---|
> > >    | *Watermark Removal* | Would you mind giving the image a fresh start by removing the watermark? | Could you perform a bit of digital alchemy and transform the image by removing that watermark? The picture deserves to be seen in its most pristine form, free from any distractions. Let the original artwork emerge unblemished, allowing every detail to shine through. Your skill in making such transformations would be greatly appreciated. |
> > >    | *Colorization* | Could you breathe life into this image by adding vibrant colors that capture its essence?  | Would you be able to transform this image by imbuing it with a rich palette of colors? Imagine each stroke of color enhancing the depth and emotion within the scene, turning the monochrome into a vivid masterpiece. Your artistic touch could reveal hidden layers, adding warmth and character to every detail. The final creation will undoubtedly captivate the eye. |
> > >    | *Dehazing* | Please lift the fog from this image, revealing its crisp and vibrant essence. | Imagine peeling back a veil to reveal the true clarity beneath—could you do the same for this image by removing the haze? The picture deserves to be seen in all its vivid glory, free from any clouding effects. Your skill in restoring sharpness would transform this image into something truly striking, with every detail standing out beautifully. |
> > >    | *Snow Removal* | Please sweep away the snowy blanket covering this image. | Could you bring the image back to its original state by removing the snow that currently veils it? The scene underneath holds a story waiting to be told, free from the cold layer above. Allow the true colors and details to shine through, unmasked by the wintry cover. Your expertise in restoring this image to its pristine form would be invaluable. |
> > >    | *Super Resolution* | Could you work your digital magic to sharpen this image and enhance its resolution? | Would you kindly apply your expertise to refine this image by eliminating the blur and boosting its resolution? The goal is to reveal the full clarity and sharpness hidden within, making every detail stand out. By enhancing its quality, you’ll allow the image to achieve its true potential, presenting it with the vividness and precision it merits. |
> > >    | *Low-light Enhancement* | Could you boost the low light in this image to reveal more detail and clarity? | Could you work your magic on this image by subtly enhancing its low-light areas? The goal is to brighten up the dim portions without altering the overall mood, allowing the hidden details to emerge while maintaining the original tone. A careful balance in lighting adjustment will ensure the image remains true to its essence, while also improving visibility and depth. Your expertise in making this enhancement would be greatly appreciated. |

---

> > > > ### Author Response · Authors · 2024-08-11
> > > >
> > > > 3. **Explanation of the HGS Design**
> > > >
> > > >     The HGS is designed to preserve crucial structures from the original image and avoid the distortion of high-frequency details, such as the text shown in Figure 5. Traditional low-level processing methods address this effectively by employing skip-connection layers in the pixel space, ensuring these details remain intact.
> > > >
> > > >     However, in Diffusion-based methods, skip-connections are only implemented in the latent space within the U-Net. Consequently, baseline methods like Diff-Plugin and MGIE, which focus solely on training the U-Net, can only retain details at a downsampled spatial dimension. When these methods perform upsampling using a frozen VAE decoder, the absence of skip-connections in the VAE leads to the distortions.
> > > >
> > > >     To address this issue, updating the U-Net alone does not work; it is essential to update the decoder as well. Additionally, Stable Diffusion operates at the latent level ($z_t$), so directly accessing pixel-level noised map ($x_t$) is not feasible.
> > > >
> > > > 4. **Clarification of the Auxiliary Prompt**
> > > >
> > > >     We would like to gently point out that the Auxiliary Prompt *is displayed in text form*, as illustrated by the blue and yellow prompts in Figure 4. Specifically, the text-based auxiliary prompt provides supplementary information through the U-Net's cross-attention layers, rather than the image encoder, as depicted in Figure 2.
> > > >
> > > >     The proposed HGS enhances image quality in most cases, as demonstrated by the improvement in LPIPS from 0.2068 to 0.1600, as shown in the General Response II table. We observed that HGS does not consistently improve performance in all scenarios, as noted in the limitations section. We will make further improvements in this direction in future work.
> > > >
> > > >     However, preserving the details of the original image remains an unresolved issue for all diffusion baselines. Our HGS method is effective in addressing this problem, as noted in your comments. We are among the first to make significant progress in tackling this issue. Therefore, we believe that our two proposed modules are both effective and distinct.
> > > >
> > > > We sincerely hope this response addresses your concerns. Please let us know if you have any further questions. Thank you again!

---

> > > > > ### Comment · Reviewer_6nKQ · 2024-08-13
> > > > >
> > > > > Thanks for the responses. I believe my previous concerns have mostly been addressed and have changed my score.

---

> ### Author Response · Authors · 2024-08-13
>
> Thank you sincerely for your engaged and increased score!
>
> Your constructive feedback has been instrumental in enhancing the clarity and contribution of our paper. The positive discussions we've had and the recognition of our efforts truly encapsulate the essence of the OpenReview process.
>
> Your time and efforts are immensely appreciated.

---

### Official Review · Reviewer_sDoY · 2024-07-05

**Soundness:** 3
**Presentation:** 3
**Contribution:** 3
**Rating:** 6
**Confidence:** 4

**Summary:**

This paper introduces PromptFix, a unified model designed to intelligently interpret and execute customized human instructions across a variety of low-level image tasks. To address the issue of spatial information loss in stable diffusion, PromptFix introduces a high-frequency guidance sampling strategy. Additionally, to tackle the degradation adaptation problem, PromptFix incorporates an auxiliary prompt module, providing models with more descriptive text prompts to enhance controllability in image generation.

**Strengths:**

1. The authors construct a comprehensive dataset tailored for low-level image processing tasks.
2. The proposed PromptFix presents a user interactive image processing method, exhibiting superior zero-shot capabilities in blind restoration and hybrid degradation tasks.
3. Both the visual and quantitative results demonstrate the effectiveness of PromptFix.
4. The paper is well-written, with the motivation, method, and experiments clearly explained.

**Weaknesses:**

1. The authors claim that when user-input instructions are discarded, PromptFix occasionally performs text-to-image generation based on the auxiliary prompt rather than image processing tasks. It is recommended that authors provide examples of these failed cases and evaluate whether the generated images can preserve the layout and structure of the input images.
2. There is a concern that PromptFix might learn a mistaken shortcut by memorizing the degradation type of the input image. Therefore, authors are encouraged to demonstrate PromptFix's ability to correctly execute tasks, such as coloring a snowy image while preserving the snow.
3. Authors assert the superior generalization and zero-shot capabilities in blind restoration and combination tasks. Therefore, it is suggested that authors evaluate the model on out-of-distribution and real-world datasets to substantiate these claims.

**Questions:**

The authors claim that the HGS strategy has the potential to introduce spatial information loss. However, the visual results in Figure 5 are insufficient to prove HGS's effectiveness in maintaining image fidelity. Quantitative results should be presented to demonstrate that the proposed HGS module is indispensable to PromptFix.

**Limitations:**

Authors claimed the limitations in the Appendix.

---

> ### Author Rebuttal · Authors · 2024-08-07
>
> Thank you for your time, thorough comments, and valuable suggestions. We are pleased that you acknowledged our clearly explained idea, the well-written paper, and our convincing experiments.
>
> > **Q1**: The authors claim that when user-input instructions are discarded, PromptFix occasionally performs text-to-image generation based on the auxiliary prompt rather than image-processing tasks. It is recommended that authors provide examples of these failed cases and evaluate whether the generated images can preserve the layout and structure of the input images.
>
>
> For this very occasional case mentioned in the limitations section, the results do not preserve structure at all. They behave like regular text-to-image generation. We will include examples in the revision.
>
>
> > **Q2**: There is a concern that PromptFix might learn a mistaken shortcut by memorizing the degradation type of the input image. Therefore, authors are encouraged to demonstrate PromptFix's ability to correctly execute tasks, such as coloring a snowy image while preserving the snow.
>
> Thank you for your advice. In the uploaded PDF, we present the case as per your suggestion to demonstrate that our method performs tasks correctly. We have placed the example in the top right corner, showing a black-and-white snowy road scene being colorized while retaining the snow.
>
>
> > **Q3**: Authors assert superior generalization and zero-shot capabilities in blind restoration and combination tasks. Therefore, it is suggested that authors evaluate the model on out-of-distribution and real-world datasets to substantiate these claims.
>
> Good suggestion. Figure 4 and several figures in the appendix show real-world processing. To further support this claim, we provide more qualitative results on real-world testing in the updated PDF. Please refer to it.
>
>
> > **Q4**: The authors claim that the HGS strategy has the potential to introduce spatial information loss. However, the visual results in Figure 5 are insufficient to prove HGS's effectiveness in maintaining image fidelity. Quantitative results should be presented to demonstrate that the proposed HGS module is indispensable to PromptFix.
>
> A quantitative ablation study is presented in the **General Response - II**. Please check them for details.

---

> ### Author Response · Authors · 2024-08-12
> **A friendly reminder**
>
> Dear Reviewer,
>
> I would like to send a kind reminder. Has our response addressed your concerns? The reviewer discussion period is nearing its end, and we eagerly await your reply. Your suggestions and comments are invaluable to the community. Thank you!
>
> Best, The authors

---

> > ### Comment · Reviewer_sDoY · 2024-08-13
> >
> > Thanks to the authors for their detailed responses. After considering the other reviews and the replies provided, I can confirm that the authors have addressed all my concerns. I raised my final rating to weak accept.

---

> > > ### Author Response · Authors · 2024-08-13
> > >
> > > As the phase of author-reviewer discussions draws to a close, we are so pleased to note your recognition of our efforts and the raised score.
> > >
> > > We are grateful for the valuable suggestions you posed and appreciate the time and effort you devoted to the review process.

---

### Official Review · Reviewer_uR5Y · 2024-07-13

**Soundness:** 3
**Presentation:** 2
**Contribution:** 3
**Rating:** 5
**Confidence:** 4

**Summary:**

This paper employs prompts to perform low-level image restoration tasks using pretrained diffusion models. To facilitate this, a substantial paired dataset with image restoration instructions was collected. The proposed method relies on latent diffusion models, incorporating the input low-quality image as an additional input and using a VLM to generate auxiliary prompts, serving as another text condition for the network. During sampling, the paper introduces High-frequency Guidance Sampling, wherein the VAE decoder is optimized to better capture the high-frequency details of the input images.

**Strengths:**

1. This paper studies an interesting problem, utilizing instructions to do low-level image restoration tasks.
2. The qualitative results look promising on various image restoration tasks.

**Weaknesses:**

1. The dataset is collected by manually performing the degradation which has a distribution gap with the real-world low-quality images. How will the method perform for the real-world image degradation task, for example, motion blur captured by the phone camera?
2. Regarding the comparison, since the baselines are not trained on the collected dataset, it might be a bit unfair as they aren't aware of the low-level restoration tasks. It would be better to compare the methods under a similar setting.

**Questions:**

1. Regarding the additional cross-attention layers, do you simply add another cross-attention layer after all the existing cross-attention layers and do the two cross-attention sequentially or parallel? And do you clone the weights as well?
2. In Alg 1, it seems that the decoder is updated during the testing time for each input? Since the loss is based on the input (low-quality image), will it affect the output image quality?
3. I'm also curious about how robust the method is for different instructions. For example, if we use different text prompts as instructions, how will the image quality change?

**Limitations:**

Yes.

---

> ### Author Rebuttal · Authors · 2024-08-07
>
> Thank you for the time, thorough comments, and nice suggestions. We are pleased to clarify your questions step-by-step.
>
> > **Q1**: The dataset is collected by manually performing the degradation which has a distribution gap with the real-world low-quality images. How will the method perform for the real-world image degradation task, for example, motion blur captured by the phone camera?
>
> 1. Our data isn't entirely human-crafted degradation. For instance, in low-light enhancement, we used 47,139 real images, with paired images taken by cameras at different ISO values.
> 2. In the updated PDF's bottom-right image, we provide how PromptFix handles natural photos with motion blur. This photo was taken with a camera.
>
>
>
> > **Q2**: Regarding the comparison, since the baselines are not trained on the collected dataset, it might be a bit unfair as they aren't aware of the low-level restoration tasks. It would be better to compare the methods under a similar setting.
>
> Good suggestion. We discuss this in the **General Response - IV**, please refer to it.
>
>
> > **Q3**: Regarding the additional cross-attention layers, do you simply add another cross-attention layer after all the existing cross-attention layers and do the two cross-attention sequentially or parallel? And do you clone the weights as well?
>
> We add cross-attention layers sequentially after the existing ones. These new layers for auxiliary prompts are initialized with the original cross-attention weights and tuned during training. We will clarify this in the revision.
>
>
> > **Q4**: In Alg 1, it seems that the decoder is updated during the testing time for each input? Since the loss is based on the input (low-quality image), will it affect the output image quality?
>
> - Yes, it will be updated for each input.
> - Our HGS may not always enhance performance. In some cases, HGS makes the restored image slightly resemble the degraded image. We have discussed this limitation in the second paragraph of Appendix A.2.
>
>
> > **Q5**: I'm also curious about how robust the method is for different instructions. For example, if we use different text prompts as instructions, how will the image quality change?
>
> Thanks for your advice. We discuss this in the **General Response - III**, please refer to it.

---

> ### Author Response · Authors · 2024-08-12
> **A friendly reminder**
>
> Dear Reviewer,
>
> I would like to send a kind reminder. Has our response addressed your concerns? The reviewer discussion period is nearing its end, and we eagerly await your reply. Your suggestions and comments are invaluable to the community. Thank you!
>
> Best, The authors

---

> ### Comment · Area_Chair_fWx7 · 2024-08-13
>
> Dear Reviewer,
>
> The authors have posted an author response here. Could you go through the rebuttal and update your opinion and rating as soon as possible?
>
> Your AC

---

> ### Comment · Reviewer_uR5Y · 2024-08-13
>
> Thanks for the response. After reading the rebuttal and other reviews, I believe my concerns have been resolved and thus would like to increase my initial score to 5.

---

> > ### Author Response · Authors · 2024-08-14
> >
> > Thank you! We’re pleased that our rebuttal has satisfactorily addressed your concerns. We greatly appreciate your suggestions for improving our paper and your positive feedback.

---

### Official Review · Reviewer_gUoB · 2024-07-22

**Soundness:** 2
**Presentation:** 2
**Contribution:** 2
**Rating:** 4
**Confidence:** 4

**Summary:**

This paper addresses low-level image processing tasks using a unified, Diffusion-based method. The key idea is to construct a dataset comprising pairs of editing instructions and targets for a variety of tasks and fine-tune a pre-trained text-to-image Diffusion Model on this dataset. Further innovations include augmenting editing instructions with text descriptions from a Vision Language Model (VLM) and a training loss that penalizes difference in high-frequency image components to facilitate detail preservation. The paper presents qualitative and quantitative results on a selected set of low-level image processing tasks, where the proposed method outperforms instruction-based Diffusion baselines and sometimes other task-specific models.

**Strengths:**

- The method addresses low-level image processing tasks in a unified framework. Specifically, the model is conditioned on text-based instructions that specify the tasks to solve. This formulation avoids training one model per task and allows knowledge (captured by model weights) to be shared among tasks.

- Low-level image processing tasks often require precise alignment of input and output pixels, yet pre-trained Diffusion models exhibit distortion as a result of lossy VAE encoding. To this end, the proposed method borrows ideas from UNet and introduces skip connections to pass along image textures that might have lost due to encoding. It further introduces a loss term to facilitate the transfer of high-frequency details. This is a sensible design that might be applicable to problem domains with similar requirements on input-output alignment.

- Strong experiment results. As the model naturally benefits from strong Diffusion priors, it performs well on tasks such as dehazing and low-light enhancement where the collection of paired training data is challenging.

**Weaknesses:**

- Lack of novelty. Instruction-based image editing using Diffusion models dates back to InstructPix2Pix, which similarly utilizes text conditioning to unify arbitrary image processing tasks. Further, I don't think it is fair to sell the dataset as a main contribution of the paper. Instruction generation using GPT-4 is not new (Both InstructionPix2Pix and LLaVA did that). The pipeline for corrupting GT images also follows standard practice. The auxiliary prompts are new but are specific to the the proposed approach. The techniques for enhancing spatial alignment are also new and seem effective, but they alone cannot justify the broad claim of the paper.

- Regarding the two proposed techniques (auxiliary prompts and loss on high-frequency components), there is no ablation study showing they are absolutely needed for the method to succeed. Qualitative and quantitative experiments are needed to show that auxiliary prompts can help in the case where input images are corrupted. Similarly, ablation experiments are needed to show how precisely the F(.) and S(.) terms in Equation 6 affects output quality.

- Some model details are lacking. For example, I am not sure I fully understand how auxiliary prompts are incorporated into the Diffusion UNet. If the cross-attention layers are replicated, are they fine-turned during training? The paper leaves me with the impression that only the LoRA convolutions are learned.

**Questions:**

- Are the experiments on low-light enhancement, dehazing, desnowing, etc. performed on simulated data? The artifacts look unrealistic to me. How does the model perform on real data? I am concerned about generalization on real data since the model is exclusively trained on simulated data.

- The paper claims in the abstract that the method "achieves comparable inference efficiency" with the baselines. Please provide concrete numbers to prove this claim.

**Limitations:**

Limitations are discussed in A.2. The model performance might be sensitive to user-provided instructions. Additionally, passing along high-frequency textures from the input image inevitably copies some artifacts to the output.

---

> ### Author Rebuttal · Authors · 2024-08-07
>
> Thanks for your constructive suggestions. Your endorsement of our method and experiments gives us significant encouragement.
>
> > **Q1.1**: Lack of novelty. Instruction-based image editing using Diffusion models dates back to InstructPix2Pix, which similarly utilizes text conditioning to unify arbitrary image processing tasks.
>
> We recognize that the InstructPix2Pix paradigm can effectively unify various image-processing tasks. However, our exploration of low-level image processing revealed several limitations:
>
> 1. **Preservation of Image Details:** InstructPix2Pix primarily focuses on image editing, often neglecting the fidelity of the original image's structure and high-frequency content. For low-level image processing tasks, preserving these details is essential. For example, when colorizing a grayscale image, losing the original high-frequency details due to VAE compression is unacceptable.
> 2. **Handling Severe Degradations:** InstructPix2Pix uses only the image and instruction as inputs, resulting in arbitrary and unrealistic outcomes for severely degraded images. An auxiliary prompt, like descriptive text, is required for better guidance.
>
> To address these issues, we propose two new mechanisms: the HGS and the Auxiliary Prompt Module. The experimental results (`strong experimental results` in your comments) demonstrate the effectiveness of our method. We believe our work provides a valuable contribution to the community.
>
> > **Q1.2**: Further, I don't think it is fair to sell the dataset as a main contribution of the paper. Instruction generation using GPT-4 is not new (Both InstructionPix2Pix and LLaVA did that). The pipeline for corrupting GT images also follows standard practice.
>
> We clarify our dataset contribution from two perspectives:
>
> 1. **The proposed dataset fills a gap**. It includes over 1 million paired images with instructions and auxiliary prompts, covering more than 7 types of low-level image processing tasks. **No such comprehensive dataset previously existed.**
> 2. **The proposed dataset is constructed beyond GPT**. We use segmentation and inpainting models to create data for object removal and creation. The bounding box and point annotations are also valuable for subject-driven generation training beyond mere image processing and editing.
>
> While the dataset construction workflow may not be highly innovative, its primary focus is on filling a gap and benefiting the community. We firmly believe that the proposed dataset is a valuable contribution to our paper, especially given the substantial resources invested in its creation.
>
>
> > **Q1.3**: The auxiliary prompts are new but are specific to the proposed approach. The techniques for enhancing spatial alignment are also new and seem effective, but they alone cannot justify the broad claim of the paper.
>
> Thank you for recognizing that our proposed auxiliary prompt and HGS are new. Our goal is to tackle the limitations of instruction-based image editing frameworks in low-level tasks by introducing two new mechanisms. Empirical results confirm the effectiveness of our proposed methods.
>
> > **Q2**: Regarding the two proposed techniques (auxiliary prompts and loss on high-frequency components), there is no ablation study showing they are absolutely needed for the method to succeed. Qualitative and quantitative experiments are needed to show that auxiliary prompts can help in the case where input images are corrupted. Similarly, ablation experiments are needed to show how precisely the F(.) and S(.) terms in Equation 6 affect output quality.
>
> We conduct experiments following your advice, please refer to **General Response - II**.
>
>
> > **Q3**: Some model details are lacking. For example, I am not sure I fully understand how auxiliary prompts are incorporated into the Diffusion UNet. If the cross-attention layers are replicated, are they fine-turned during training?
>
> The cross-attention layers for auxiliary prompting are additional, structured the same way, initialized with the original cross-attention weights, and need joint tuning during training. We will clarify this detail in our revision.
>
>
> > **Q4**: Are the experiments on low-light enhancement, dehazing, desnowing, etc. performed on simulated data? The artifacts look unrealistic to me. How does the model perform on real data? I am concerned about generalization on real data since the model is exclusively trained on simulated data.
>
> Our curated dataset includes realistic data. For the desnowing task, we used 2329 real data from datasets [1-2]; for the dehazing task, we used 5422 real data from datasets [2-5]; for the low-light enhancement task, we used 47139 real data from datasets [6-9], and these low-light image pairs are both taken by cameras with different ISO values.
>
> Besides, we provide real-world degraded image processing examples in the uploaded PDF to demonstrate the generalization ability of our model on real data. Please refer to it.
>
> [1] Desnownet: Context-aware deep network for snow removal. TIP (2018)
>
> [2] Jstasr: Joint size and transparencyaware snow removal algorithm based on modified partial convolution and veiling effect removal. ECCV (2020)
>
> [3] Benchmarking single-image dehazing and beyond. TIP (2018)
>
> [4] Dense-haze: A benchmark for image dehazing with dense-haze and haze-free images. ICIP (2019)
>
> [5] O-haze: a dehazing benchmark with real hazy and haze-free outdoor images. CVPRW (2018)
>
> [6] Deep retinex decomposition for low-light enhancement. BMVC (2018)
>
> [7] Learning to see in the dark. CVPR (2018)
>
> [8] Seeing motion in the dark. ICCV (2019)
>
> [9] Seeing dynamic scene in the dark: A high-quality video dataset with mechatronic alignment. ICCV (2021)
>
>
> > **Q5**: The paper claims in the abstract that the method "achieves comparable inference efficiency" with the baselines. Please provide concrete numbers to prove this claim.
>
> We provide a comparison of different models' FLOPs in Appendix A.4 to support this claim. Please refer to this section.

---

> ### Author Response · Authors · 2024-08-12
> **A friendly reminder**
>
> Dear Reviewer,
>
> I would like to send a kind reminder. Has our response addressed your concerns? The reviewer discussion period is nearing its end, and we eagerly await your reply. Your suggestions and comments are invaluable to the community. Thank you!
>
> Best, The authors

---

> ### Comment · Area_Chair_fWx7 · 2024-08-13
>
> Dear Reviewer,
>
> The authors have posted an author response here. Could you go through the rebuttal and update your opinion and rating as soon as possible?
>
> Your AC

---

### Author Rebuttal · Authors · 2024-08-07

# General Response to Reviewers and ACs

We thank the reviewers for their detailed and valuable comments. To better support our response, we have uploaded a rebuttal PDF (need to download it) containing the supporting materials. The figures within this PDF are labeled using Roman numerals, such as Figure A.

In this post:

- (1) We summarize positive feedback from the reviews.
- (2) We address four common issues raised in the reviews.

## (1) Positive feedbacks

- **Strong empirical performance of the proposed method**
    - `[gUoB]`: "*Strong experiment results. ... it performs well on tasks such as dehazing and low-light enhancement*"
    - `[uR5Y]`: "*The qualitative results look promising on various image restoration tasks.*"
    - `[sDoY]`: "*Both the visual and quantitative results demonstrate the effectiveness of PromptFix.*"
    - `[6nKQ]`: "*Experiments show competitive performance on various restoration tasks.*"

- **The proposed dataset**
    - `[6nKQ]`: "*The large-scale, instruction-following dataset that covers comprehensive image-processing tasks could be helpful in the field of low-level image processing.*"
    - `[sDoY]`: "*The authors construct a comprehensive dataset tailored for low-level image processing tasks.*"

- **Presentation**
    - `[sDoY]`: "*The paper is well-written, with the motivation, method, and experiments clearly explained.*"


## (2) Addressing common issues

### **I**: Real-world Testing

Reviewers`[gUoB]`, `[uR5Y]`, `[sDoY]`, and`[6nKQ]` noted that more real-world data testing is expected. We have included results in Figure 4 and multiple cases in the appendix to demonstrate real-world image processing performance. To further illustrate our model's robustness, we have added more qualitative results using real-world low-level images in the uploaded PDF. Please refer to it.

### **II**: Quantitative Study on HGS and Auxiliary Prompting

Reviewers `[gUoB]`, `[sDoY]`, and `[6nKQ]` suggest using numeric metrics to evaluate the importance of our HGS and Auxiliary Prompt Module for the proposed PromptFix. We conducted quantitative experiments, as shown in the table below.

| HGS |   | Auxiliary Prompting | LPIPS↓  | ManIQA↑  |
| ---: | :--- | :---: | --- | --- |
| $\mathcal{F}(\cdot)$ |  $\mathcal{S}(\cdot)$ |  |  |  |
|  |  | $\checkmark$ | 0.2068 | 0.6487 |
|  | $\checkmark$ | $\checkmark$ | 0.1707 | 0.6300 |
| $\checkmark$ |  | $\checkmark$ | 0.1795 | 0.6195 |
| $\checkmark$ | $\checkmark$ |  | 0.1990 | 0.5856 |
| $\checkmark$ | $\checkmark$ | $\checkmark$ | 0.1600 | 0.6274 |

In the table, $\mathcal{F}(\cdot)$ and $\mathcal{S}(\cdot)$ represent the type of high-frequency operators used in HGS to guide sampling. These quantitative results indicate the effectiveness of the proposed HGS and auxiliary prompting and their essential role in PromptFix.
In addition to the results above, Tables 2 and 3 in the paper also demonstrate the superiority of the Auxiliary Prompt Module in blind restoration and multi-task processing.


### **III**: Ablation Study on Different Types of Instruction Prompt

Reviewers `[uR5Y]` and `[6nKQ]` suggest we assess the model's generalization to various human instructions. To verify this, we conduct ablation comparisons with three types of prompts:
- $\mathbb{A}$: instructions used during training;
- $\mathbb{B}$: out-of-training human instructions with fewer than 20 words;
- $\mathbb{C}$: out-of-training human instructions with 40-70 words.

Instructions $\mathbb{B}$ and $\mathbb{C}$ are generated by GPT-4. The experimental results are presented in the following table:

| Instruction Type | LPIPS↓ | ManIQA↑ |
| --- | --- | --- |
| $\mathbb{A}$ | 0.1600 | 0.6274 |
| $\mathbb{B}$ | 0.1639 | 0.6258 |
| $\mathbb{C}$ | 0.1823 | 0.5958 |

The model's performance slightly declines with out-of-training instructions, but the change is negligible. This indicates that our model is robust for instructions under 20 words, which is generally sufficient for low-level processing tasks.
We observe a performance drop with longer instructions, possibly due to the long-tail effect of instruction lengths in the training data. Although low-level processing tasks usually don't require long instructions, addressing this issue by augmenting the dataset with longer instructions could be a direction for future work.


### **IV**: Comparison with the Baseline Finetuned on Our Dataset

Reviewers `[uR5Y]` and `[6nKQ]` note that some baseline models may not trained with our proposed dataset. To ensure the effectiveness of PromptFix and improve the fairness of our comparison, we initialize the pre-trained checkpoint of instructdiff and fine-tune it on our dataset for about 150,000 iterations using a learning rate of 5e-6 on 16 80G GPUs. We refer to this as InstructDiff\*. The table below shows the detailed quantitative results:

| Method | LPIPS↓ | ManIQA↑ |
| --- | --- | --- |
| InstructDiff | 0.2815 | 0.5560 |
| InstructDiff\* | 0.2149 | 0.6086 |
| Ours | 0.1600 | 0.6274 |


> The results of the above three quantitative results consolidate all low-level processing tasks. We will update these experimental analyses and include real-world test visualizations in the revision.

---

### Decision · Program_Chairs · 2024-09-25

**Decision:**

Accept (poster)

**Comment:**

After the rebuttal, the paper received the ratings of 4/5/6/6. Basically mostly all the concerns are addressed by the rebuttal provided by the authors. After examining the reviews and the rebuttal for the reviewer with a rating of 4, the AC believes the concerns should be addressed as well, as these concerns are the common ones shared by other reviewers. Generally, the reviewers recognize the strengths of the paper: a constructed dataset for image editing with prompts, PromptFix provides an interactive approach for image editing, and the experiments are compressive and convincing. Thus, the AC recommends acceptance of the paper. The authors are recommended to incorporate the suggestions from the reviewers for the camera-ready version.